# PNPLA7 mediates Parkin-mitochondrial recruitment in adipose tissue for mitophagy and inhibits browning

Xuetao Ji[1,11], Xu Zhang[1,11], Tong Zhang[1], Yao Xue[1], Mengping He[1], Chaopu Li[1], Yun Huang[1], Haoyu Wang[1], Jing Ju[1], Li'e Cai[1], Yuzhu Wang[1], Ning Wang[1], Lijuan Fan[1], Hui Tong[1], Heng Fan[2], Qinsheng Chen[3], Qinwei Lu[3], Cong Li[3], Huiru Tang [3], Yongsheng Chang [4], Xingxing Kong [5], Hanming Shen [6], Aihua Gu [1], Hui Liang[7], Hongwen Zhou[1,8], Qian Wang[1] ✉ & John Zhong Li [1,9,10] ✉

PINK1/Parkin-mediated ubiquitin-dependent mitophagy is a critical negative regulatory machinery for browning in the inguinal white adipose tissue (iWAT). However, the precise regulatory mechanism underlying PINK1/Parkin-mediated mitophagy during browning of iWAT remains largely unknown. Here we report that PNPLA7, an Endoplasmic Reticulum and mitochondria-associated membrane (MAM) protein, inhibits browning of iWAT by promoting PINK1/Parkin-mediated mitophagy upon cold challenge or β3-adrenergic receptor agonist treatment. With genetic manipulation in mice, we show that adipose tissue overexpressing PNPLA7 induces mitophagy, abolishes iWAT browning and interrupts adaptive thermogenesis. Conversely, conditional ablation of PNPLA7 in adipose tissue promotes browning of iWAT, resulting in enhanced adaptive thermogenesis. Mechanistically, PNPLA7 interacts with Parkin to promote mitochondrial recruitment of Parkin for mitophagy activation and mitochondria degradation by disrupting PKA-induced phosphorylation of Parkin under cold challenge. Taken together, our findings suggest that PNPLA7 is a critical regulator of mitophagy that resists cold-induced browning of iWAT, thus providing a direct mechanistic link between mitophagy and browning of iWAT.

Brown adipose tissue (BAT) protects against hypothermia by its vast expression of mitochondrial uncoupling protein 1 (UCP1), which drives adaptive non-shivering thermogenesis at low ambient temperature in mammals[1,2]. In addition to BAT, non-shivering thermogenesis is also generated by UCP1+ beige adipocytes within the white adipose depots, particularly in the subcutaneous inguinal white adipose tissue (iWAT) of mice[3–5]. Beige adipocytes are strongly induced in response to chronic cold exposure or adrenergic signaling, exhibiting a remarkable reprogramming from white adipocytes through browning[6–10], this browning process has been suggested to have strong anti-obesity and

anti-diabetic benefits and represents a promising strategy to counteract obesity and metabolic dysfunctions[10–17].

The reprogramming from white to beige adipocyte is critically governed through the regulation of mitochondrial number and the levels of thermogenic machinery's components under different external stimuli. Mitochondrial homeostasis is regulated by a balance between mitochondrial biosynthesis and degradation[18–20]. Mitophagy, a selective degradation of dysfunctional mitochondria, is a quality control mechanism that is imperative to the maintenance of a healthy mitochondrial network[21,22]. Emerging evidences suggest that

mitophagy plays a key role in brown and beige adipocyte mitochondrial homeostasis during cold adaptation[23–27]. Inhibition of mitophagy increases thermogenic activity in the beige adipocytes[28]. Studies of mice subjected to invalidation of key factors in mitophagy indicate that the dampened induction of browning in these mice is the result of mitophagy impairment in adipocytes[29,30]. PINK1/Parkin-mediated mitophagy is a widely characterized ubiquitin-dependent mitophagy pathway[31,32], while NIX/BNIP3L and FUNDC1 contribute to Receptor-mediated mitophagy[33–36]. Recent studies have shown that Parkin-mediated mitophagy is downregulated in iWAT browning and is essential for beige to white adipocytes transition (whitening) after the withdrawal of external stimuli[27,37,38]. Parkin deletion specifically in adipose tissue protects mice against high-fat diet and aging-induced obesity by coordinating mitophagy with mitochondrial biogenesis in white adipocytes[24,30]. In addition, activation of β3 adrenergic receptor (β3-AR) by norepinephrine has been shown to inhibit Parkin mitochondrial recruitment through PKA-mediated phosphorylation, thereby reducing mitophagy and promoting browning of WAT[26,39]. However, the molecular events by which Parkin is recruited to mitochondria in WAT has not been fully elucidated.

PNPLA7 belongs to the PNPLA family and is highly expressed in metabolically active tissues such as liver, adipose tissue, and skeletal muscle[40]. The expression of PNPLA7 is down-regulated in hepatocellular carcinoma (HCC) cell lines and highly responsive to nutritional condition in 3T3-L1 cells[40,41]. PNPLA7 has been identified as an endoplasmic reticulum (ER) transmembrane protein that functions as lysophosphatidylcholine hydrolase and interacts with lipid droplets through its catalytic domain[42]. We previously reported that PNPLA7 regulates hepatic VLDL secretion by modulating ApoE stability through protein-protein interaction, independent of its catalytic activity[43]. A recent study indicates that PNPLA7 also controls hepatic choline and methionine metabolism, and PNPLA7 deficiency can lead to marked reduction in fat mass and increases adipocyte browning[44]. Although PNPLA7 regulates nutrient and lipid metabolism in the liver, its physiological role in adipose tissue remains unknown.

In this study, we found that PNPLA7 is an ER and mitochondrial-associated membrane (MAM) protein. The expression of PNPLA7 is reduced by cold exposure or β3-AR agonist treatment and inversely correlated with the expression levels of thermogenic protein UCP1 in the adipose tissue. In addition, overexpression of PNPLA7 in adipose tissue was found to increase mitochondrial recruitment of Parkin, leading to augmented mitophagy and mitochondrial degradation even under cold conditions, whereas the ablation of Pnpla7 in adipose tissue reduces mitophagy and promotes iWAT browning and adaptive thermogenesis. We further revealed that PNPLA7 interacts with Parkin through its C-terminal region, consisting of the Patatin domain, and blocks PKA-mediated phosphorylation of Parkin. These events facilitate Parkin recruitment to mitochondria to promote mitophagy for mitochondria degradation in iWAT under cold stress. Therefore, our study suggests that PNPLA7-mediated Parkin's mitochondrial recruitment and mitophagy is a critical regulatory module to control iWAT browning and provide a potential pharmaceutical target to ameliorate obesity and related metabolic disorders.

## Results

### PNPLA7 is downregulated in adipose tissues during browning
Recently, others have reported Pnpla7 is highly expressed in various mouse tissues, including the liver, skeletal muscles, and testes, while lower mRNA expression was observed in the brown and white fat tissue[44]. To determine the role of PNPLA7, if any, in the adipose tissues, we first assessed PNPLA7 protein expression level in the interscapular BAT (iBAT), subcutaneous inguinal WAT (iWAT) and epididymal WAT (eWAT) in mice. Western blotting showed that PNPLA7 proteins were more abundant in the BAT and iWAT compared to eWAT (Fig. 1a). However, in mice under adaptive thermogenesis induced by either

cold exposure (Fig. 1b, c) or β3-AR agonist (CL316,243) treatment (Fig. 1d, e), both Pnpla7 mRNA and protein expression were reduced, strikingly contrasted to the potent activation of the thermogenic markers Pgc-1α and Ucp1 in both iWAT and BAT. To validate this intriguing observation, we treated primary adipocytes derived from iWAT with different browning stimuli functioning through various mechanisms, including norepinephrine (Fig. 1f), CL 316,243 (Fig. 1g), isoproterenol (Fig. 1h), dibutyryl-cAMP (Fig. 1i), triiodothyronine (Fig. 1j), and Rosiglitazone (Fig. 1k), and detected a consistent downregulation of PNPLA7, implicating this is probably a common mechanism underlying adaptive thermogenesis.

### PNPLA7 is a negative regulator of browning in iWAT
Having revealed this inverse correlation between adipose PNPLA7 expression and thermogenic response both in vivo and in vitro, we hypothesized that PNPLA7 is a negative regulator of catabolic function in adipose tissue. To this end, we generated two transgenic mouse lines that specifically overexpressed PNPLA7 in adipose tissues (Pnpla7Tg mice) (Supplementary Fig. 1a–g) and performed indirect calorimetry. Our analysis indicates that Pnpla7Tg mice with PNPLA7 overexpression exhibit reduced oxygen consumption and energy expenditure following β3-AR agonist (CL 316,243) administration compared to wild type mice (Fig. 2a, b), independent of physical activity and food intake (Supplementary Fig. 1h, i). Since BAT plays an important role for adaptive thermogenesis in response to cold exposure in mice, the effect of PNPLA7 overexpression in BAT was examined using comprehensive approaches, including H&E staining (Supplementary Fig. 1j), immunohistochemical (Supplementary Fig. 1k), transmission electron microscopic (TEM) (Supplementary Fig. 1l) and western blotting analyses (Supplementary Fig. 1m, n). All the experiments revealed that the morphological feature and mitochondrial ultrastructure of BAT as well as the thermogenic and mitochondrial OXPHOS proteins level in the BAT of Pnpla7Tg mice were not affected. These results suggest that the impaired thermogenic capacity in Pnpla7Tg mice is likely not attributable to BAT dysfunction.

We then shifted to investigate whether overexpression of PNPLA7 could affect the browning of WAT under prolonged cold exposure or β3-AR agonist stimulation. Under room temperature (RT), the adiposity index and fat pad weight of Pnpla7Tg mice is comparable to that of the wild type mice. Expectedly, upon 7 days of cold exposure or β3-AR agonist treatment, both the adiposity index and fat pad weight decreased in wild type and Pnpla7Tg mice, however, the decrement observed in Pnpla7Tg mice is less than that of wild type mice (Fig. 2c, d). Similar observation was made for triacylglycerol (TAG) level in iWAT (Fig. 2e). Concomitantly, Pnpla7Tg mice exhibited fewer BAT-like multilocular adipocytes and fewer UCP1+ beige adipocytes in their iWAT (Fig. 2f and Supplementary Fig. 1o). Furthermore, the induction of UCP1 in the iWAT of Pnpla7Tg mice diminished upon cold exposure (Fig. 2g and Supplementary Fig. 1p).

Next, we generated a complementary adipocytes-specific-Pnpla7 knockout mice model (Pnpla7AKO) using Cre-loxp system to validate the role of PNPLA7 in iWAT browning (Supplementary Fig. 2a, b). Conversely, we found that Pnpla7AKO mice exhibit higher oxygen consumption and energy expenditure which are independent of their physical activity and food intake after β3-AR agonist (CL 316,243) injection (Fig. 2h, i and Supplementary Fig. 2c, d). Moreover, there is no notable difference in BAT morphological feature and mitochondrial ultrastructure. In addition, the thermogenic and mitochondrial OXPHOS proteins level in BAT remained unaffected (Supplementary Fig. 2e–h) in Pnpla7AKO mice. In contrast to Pnpla7Tg mice, the adiposity index, fat pad weight and TAG levels in iWAT of Pnpla7AKO mice were lower than the wild type mice under cold exposure or β3-AR agonist administration (Fig. 2j–l). In addition, these Pnpla7AKO mice exhibited

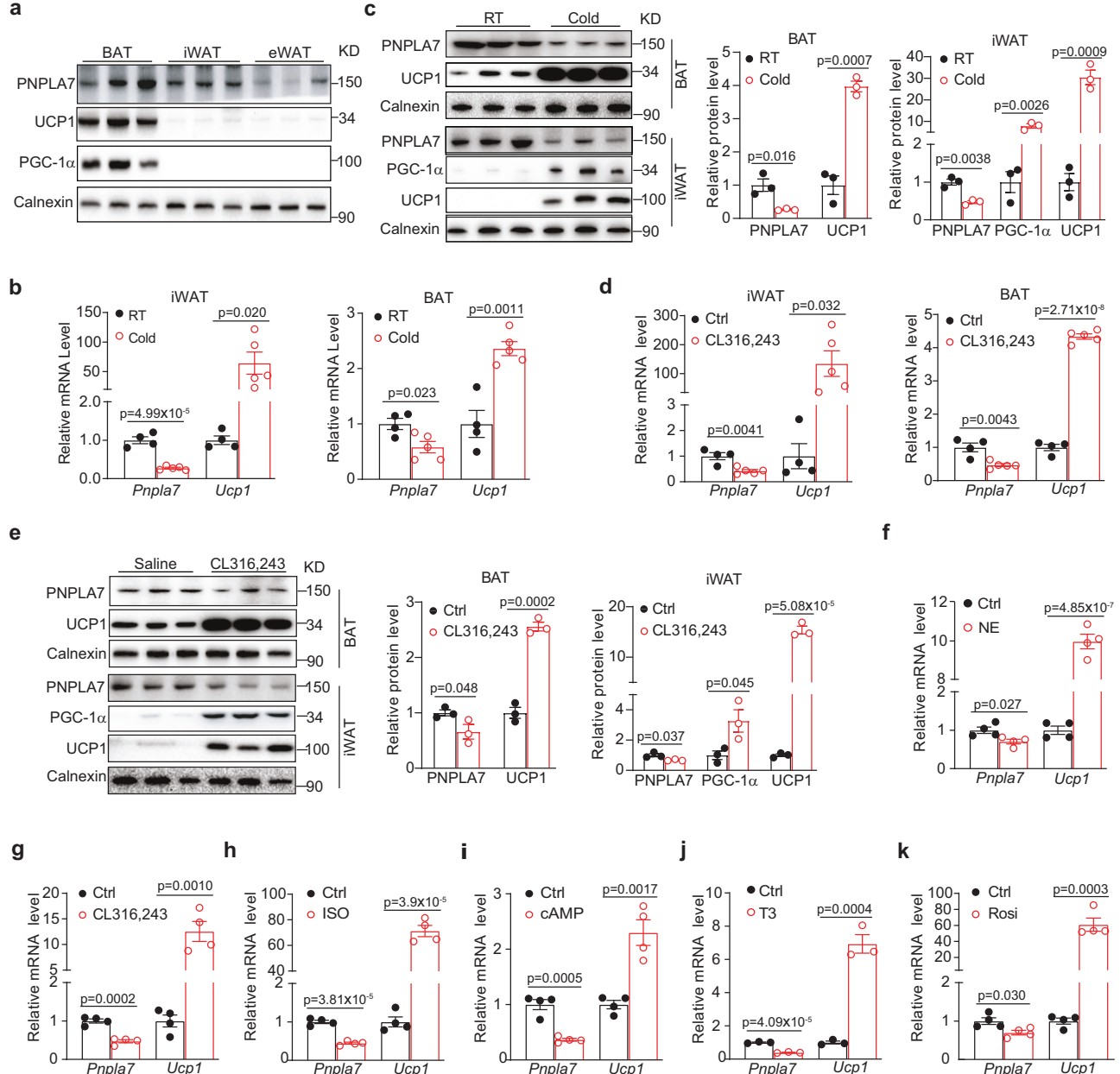

**Fig. 1 | PNPLA7 is downregulated in adipose tissues during browning.**
**a** Representative Immunoblot results of the indicated proteins in BAT, iWAT, and eWAT of 10-week-old WT *C57BL/6J* male mice (*n* = 3/group). **b** Quantitative PCR analysis of *Pnpla7* and *Ucp1* mRNA expression in iWAT and BAT of 10-week-old WT *C57BL/6J* male mice (RT: *n* = 4; Cold: *n* = 5). RT: room temperature. Data are presented as mean ± SEM. (Two-tailed Student's *t* test for 2-group comparisons). **c** Representative Immunoblot results and densitometry analysis of the indicated proteins in iWAT and BAT of 10-week-old WT *C57BL/6J* male mice (*n* = 3/group). Data are presented as mean ± SEM. (Two-tailed Student's *t* test for 2-group comparisons). **d** Quantitative PCR analysis of *Pnpla7* and *Ucp1* mRNA expression in iWAT and BAT of 10-week-old WT *C57BL/6J* male mice (Ctrl: *n* = 4; CL316,243: *n* = 5). Data are presented as mean ± SEM. (Two-tailed Student's *t* test for 2-group

comparisons). **e** Representative Immunoblot results and densitometry analysis of the indicated proteins in iWAT and BAT of 10-week-old WT *C57BL/6J* male mice (*n* = 3/group). Data are presented as mean ± SEM. (Two-tailed Student's *t* test for 2-group comparisons). **f–k** Quantitative PCR analysis of *Pnpla7* and *Ucp1* mRNA expression in differentiated primary SVF-derived adipocytes that were stimulated with vehicle (Ctrl), or 0.2 μM norepinephrine (NE) for 2 days (*n* = 4/group) (**f**), 0.1 μM CL 316,243 (CL) for 24 h (*n* = 4/group)(**g**), 10 μM isoproterenol (ISO) for 4 h (*n* = 4/group) (**h**), 500 μM dibutyryl-cAMP (cAMP) for 6 h (*n* = 4/group) (**i**), 1 μM triiodothyronine (T3) for 20 h (*n* = 3/group) (**j**), 1 μM rosiglitazone (Rosi) for 2 days (*n* = 4/group) (**k**). Data are presented as mean ± SEM. (Two-tailed Student's *t* test for 2-group comparisons). RT: room temperature. Source data are provided as a Source Data file.

more multilocular adipocytes and more abundant UCP1⁺ beige adipocytes in iWAT relative to their control littermates (Fig. 2m). In addition, an enhanced induction of UCP1 expression was also observed (Fig. 2n).

Taken together, these findings suggest that PNPLA7 is a negative regulator of adaptive thermogenesis by inhibiting browning of iWAT in response to external stimulation in the mice.

## PNPLA7 promotes mitochondrial degradation in iWAT
To understand the molecular basis of PNPLA7 in regulating iWAT browning during cold exposure, we examined the mRNA levels of various genes associated with mitochondrial biogenesis, thermogenesis, mitochondrial OXPHOS, TAG lipolysis, and fatty acid oxidation in the iWAT of both *Pnpla7^Tg^* and *Pnpla7^AKO^* mice. Here, no significant difference was observed for the indicated gene expression

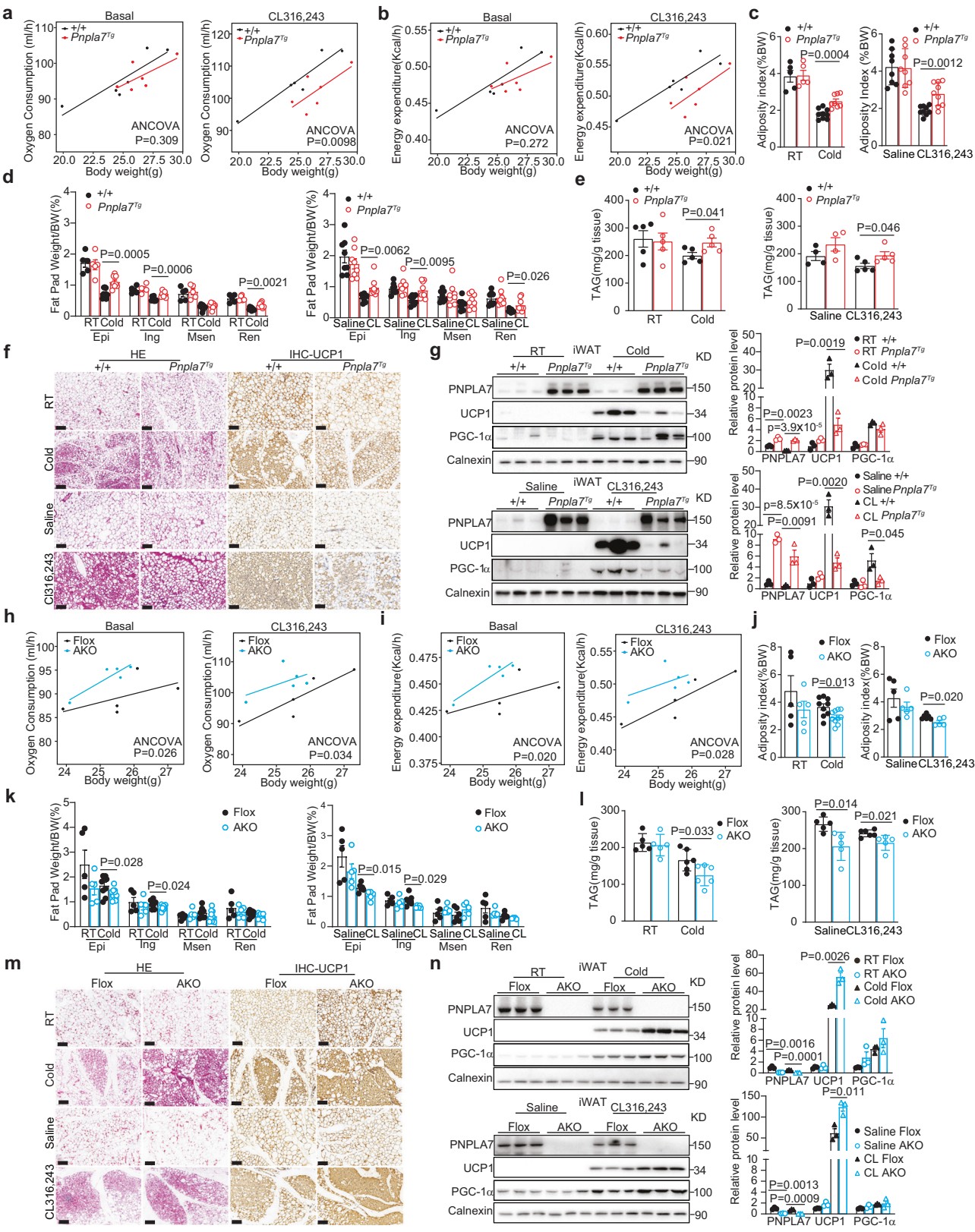

in iWAT of *Pnpla7^Tg* or *Pnpla7^AKO* mice compared with that of control mice (Supplementary Fig. 3a, b). In accordance with the lipolytic gene expression, such as *Atgl* and hormone-sensitive lipase (*Hsl*), no difference in isoproterenol stimulated glycerol release was observed in iWAT isolated from *Pnpla7^Tg* or *Pnpla7^AKO* mice (Supplementary Fig. 4a, b). In addition, manipulation of PNPLA7 expression in the adipose tissues did not affect the serum TAG, free fatty acid and

glycerol in the mice with β3-AR agonist administration (Supplementary Fig. 4c, d).

Previously, PNPLA7 was reported to possess lysophospholipase activity in COS7 cells and hepatocytes[42,44]. To explore whether the lipase activity of PNPLA7 is required for the browning of iWAT, we performed lipidomic analysis to determine the levels of the phospholipids and lysophospholipids in iWAT of *Pnpla7^Tg* and *Pnpla7^AKO*

**Fig. 2 | PNPLA7 inhibits browning of iWAT in response to cold exposure.**
**a**, **b** Regression-based analysis of absolute oxygen consumption (**a**) and energy expenditure (**b**) against body mass from 8-week-old control and *Pnpla7*[Tg] male mice. (*n* = 6/group). Oxygen consumption and energy expenditure as dependent variable, genotype as fixed variable, and body mass as covariate. (Two-sided analysis of covariance). **c**, **d** Adiposity index (**c**) and fat mass to body weight ratio (**d**) analysis of control and *Pnpla7*[Tg] mice. (RT: *n* = 5/group; cold: +/+: *n* = 8; *Pnpla7*[Tg]: *n* = 9; saline: *n* = 8/group; CL 316,243: *n* = 9/group). Data are presented as mean ± SEM. (Two-tailed Student's *t* test for 2-group comparisons). **e** TAG levels of iWAT obtained from control and *Pnpla7*[Tg] male mice (RT: *n* = 5/group; cold: *n* = 5/group; saline: *n* = 4/group; CL 316,243: *n* = 5/group). Data are presented as mean ± SEM. (Two-tailed Student's *t* test for 2-group comparisons). **f**, **g** Representative H&E and UCP1 immunohistochemical staining images (**f**) as well as representative immunoblot results and densitometry analysis (**g**) of iWAT from control and *Pnpla7*[Tg] male mice (*n* = 3 biological replicates, Scale bar = 100 μm). Data are presented as mean ± SEM. (Two-tailed Student's *t* test for 2-group comparisons). **h**, **i** Regression-based analysis of absolute oxygen consumption (**h**) and energy expenditure (**i**) against body mass from 8-week-old control and *Pnpla7*[AKO] male mice (*n* = 5/group). Oxygen consumption and energy expenditure as dependent variable, genotype as fixed variable, and body mass as covariate. (Two-sided analysis of covariance). **j**, **k** Adiposity index (**j**) and fat mass (**k**) to body weight ratio analysis of Flox and *Pnpla7*[AKO] male mice (RT: *n* = 5/group; cold: Flox: *n* = 9; AKO: *n* = 10; saline: *n* = 5/group; CL 316,243: Flox: *n* = 6; AKO: *n* = 5). Data are presented as mean ± SEM. (Two-tailed Student's *t* test for 2-group comparisons). **l** TAG levels of iWAT harvested from Flox and *Pnpla7*[AKO] male mice (RT: *n* = 5/group; cold: *n* = 6/group; saline: *n* = 5/group; CL 316,243: *n* = 5/group). Data are presented as mean ± SEM. (Two-tailed Student's *t* test for 2-group comparisons). **m**, **n** Representative H&E and UCP1 immunohistochemical staining images (**m**) as well as representative immunoblot results and densitometry analysis (**n**) of iWAT from Flox and *Pnpla7*[AKO] male mice (*n* = 3 biological replicates, scale bar = 100 μm). Data are presented as mean ± SEM. (Two-tailed Student's *t* test for 2-group comparisons). Source data are provided as a Source Data file.

mice (Supplementary Fig. 5a, b). Interestingly, no difference was observed for total levels of phosphatidylcholine (PC), lysophosphatidylcholine (LPC), phosphatidylethanolamine (PE), lysophosphatidylethanolamine (LPE), phosphatidylserine (PS), or lysophosphatidylserine (LPS) in iWAT of these mice either under RT or prolonged cold exposure. Moreover, further analysis also revealed no difference between various species of PC, LPC, PE, and LPE with mono- or polyunsaturated fatty acids in iWAT of different genotypes (Supplementary Fig. 5c–h). Together, these data suggest that the lysophospholipase activity of PNPLA7 is probably not involved in browning of iWAT.

Next, we assessed the effect of PNPLA7 protein expression on mitochondria during browning. Here, TEM images of iWAT mitochondrial ultrastructure from mice subjected to cold challenge for 7 days were collected. Interestingly, in the iWAT of *Pnpla7*[Tg] mice, abnormal mitochondrial morphology with dysmorphic cristae architecture and significant mitophagosome structure in the iWAT of *Pnpla7*[Tg] mice was observed (Fig. 3a), indicating the presence of mitophagy in the iWAT of *Pnpla7*[Tg] mice. In addition, unlike the striking increase in the number of mitochondria and cristae in the iWAT of cold-exposed wild type mice, no increment was observed in the iWAT of *Pnpla7*[Tg] mice (Fig. 3b). This was accompanied by a clear reduction in TOM20[+] beige adipocytes as shown with IHC staining (Fig. 3c) and lower mitochondrial DNA (mtDNA) content in the iWAT of *Pnpla7*[Tg] mice (Fig. 3d). In contrast, we observed a significantly increased in mitochondrial number and cristae density (Fig. 3e, f) as well as TOM20[+] beige adipocytes (Fig. 3g) and mtDNA content (Fig. 3h) in iWAT of *Pnpla7*[AKO] mice under cold exposure.

Importantly, there was no induction of mitochondrial fusion proteins mitofusin 1 (MFN1) and 2 (MFN2), TOM20 and mitochondrial OXPHOS proteins in the iWAT of *Pnpla7*[Tg] mice compared with their littermate controls under prolonged cold exposure (Fig. 3i). Conversely, these proteins expression was induced strongly in the iWAT of *Pnpla7*[AKO] mice under the same condition (Fig. 3j). Next, we further performed seahorse assay to directly measure mitochondrial respiration function in adipocytes that differentiated from SVF cells of iWAT isolated from *Pnpla7*[Tg] and *Pnpla7*[AKO] mice treated with or without browning agent Triiodothyronine (T3), an essential hormone that plays a crucial role in initiating the differentiation of SVF cells into beige adipocytes[45]. The results demonstrated that overexpression or ablation of PNPLA7 has no significant effect on the mitochondrial respiration function in adipocytes compared with that of wild-type control in the absence of T3 treatment (Fig. 3k, l). However, with T3 treatment, PNPLA7 overexpression significantly reduced basal, proton leak-linked and maximal cellular oxygen consumption rates (OCR) in *Pnpla7*[Tg] adipocytes mitochondria respiration (Fig. 3k). In contrast, ablation of PNPLA7 enhanced the mitochondrial spare respiratory capacity in T3-induced adipocytes isolated form iWAT of *Pnpla7*[AKO] mice (Fig. 3l).

To further confirm the role of PNPLA7 in white adipose tissue (WAT) during browning, we specifically introduced PNPLA7 into iWAT of wild type mice using adenoviral system via orthotopic injection (Ad-Adipo-PNPLA7). Consistently, as observed in *Pnpla7*[Tg] mice, *Ad-Pnpla7* mice exhibit reduced oxygen consumption and energy expenditure with no differences in physical activity or food intake after β3-AR agonist (CL 316,243) injection (Supplementary Fig. 6a–d). Moreover, a higher fat mass and TAG content (Supplementary Fig. 6e, f) with lower intensity of UCP1[+] and TOM20[+] beige adipocyte were observed in iWAT of *Ad-Pnpla7* mice under chronic cold exposure (Supplementary Fig. 6g). Western blot analysis also showed diminished UCP1 and mitochondrial protein levels in iWAT of these mice (Supplementary Fig. 6h, i) while mRNA expression level remain unchanged (Supplementary Fig. 6j–l). Similar to *Pnpla7*[Tg] mice, *Ad-Pnpla7* mice demonstrated lower mtDNA levels in iWAT relative to the control group (Supplementary Fig. 6m). Thus, these results further consolidate the role of PNPLA7 in iWAT.

Taken together, these data suggest that the inhibitory effect of PNPLA7 on iWAT browning is achieved by promoting mitochondrial degradation rather than inhibiting mitochondrial biogenesis.

## PNPLA7 promotes mitophagy to enhance mitochondrial degradation in iWAT

Since mitophagy has been shown to play a key role in mitochondrial degradation in beige adipose tissue, we further performed mito-Keima assay to study the possible regulatory roles of PNPLA7 on mitophagy in isolated and differentiated white adipocytes from PNPLA7-deficient (KO) or *Pnpla7*[Tg] mice. Upon treatment with DMSO, limited red fluorescence was detected in both mito-Keima expressing control adipocyters and PNPLA7 deficient adipocytes. Expectedly, in response to CCCP, a mitophagy inducer, a remarkable increase in red fluorescence was detected in the control cells. In contrast, this increase was blunted in the PNPLA7-deficient adipocytes (Fig. 4a, b). Moreover, ablation of PNPLA7 significantly blocked CCCP-induced mitophagosomes formation (Fig. 4c, d) as well as the degradation of mtDNA (Fig. 4e). Consistently, higher level of mitochondrial outer membrane, inner membrane and matrix proteins were detected in PNPLA7 deficient adipocytes with CCCP or valinomycin treatment, respectively (Fig. 4f, g).

In contrast, an increase in red fluorescence intensity in mito-Keima assay and mitophagosomes formation as well as enhanced mtDNA and mitochondrial protein degradation were observed in adipocytes overexpressing PNPLA7 after treatment with CCCP or Valinomycin (Supplementary Fig. 7a–f).

Taken together, these data indicate that PNPLA7 inhibits iWAT browning in response to external stimuli by promoting mitophagy and enhances mitochondrial degradation.

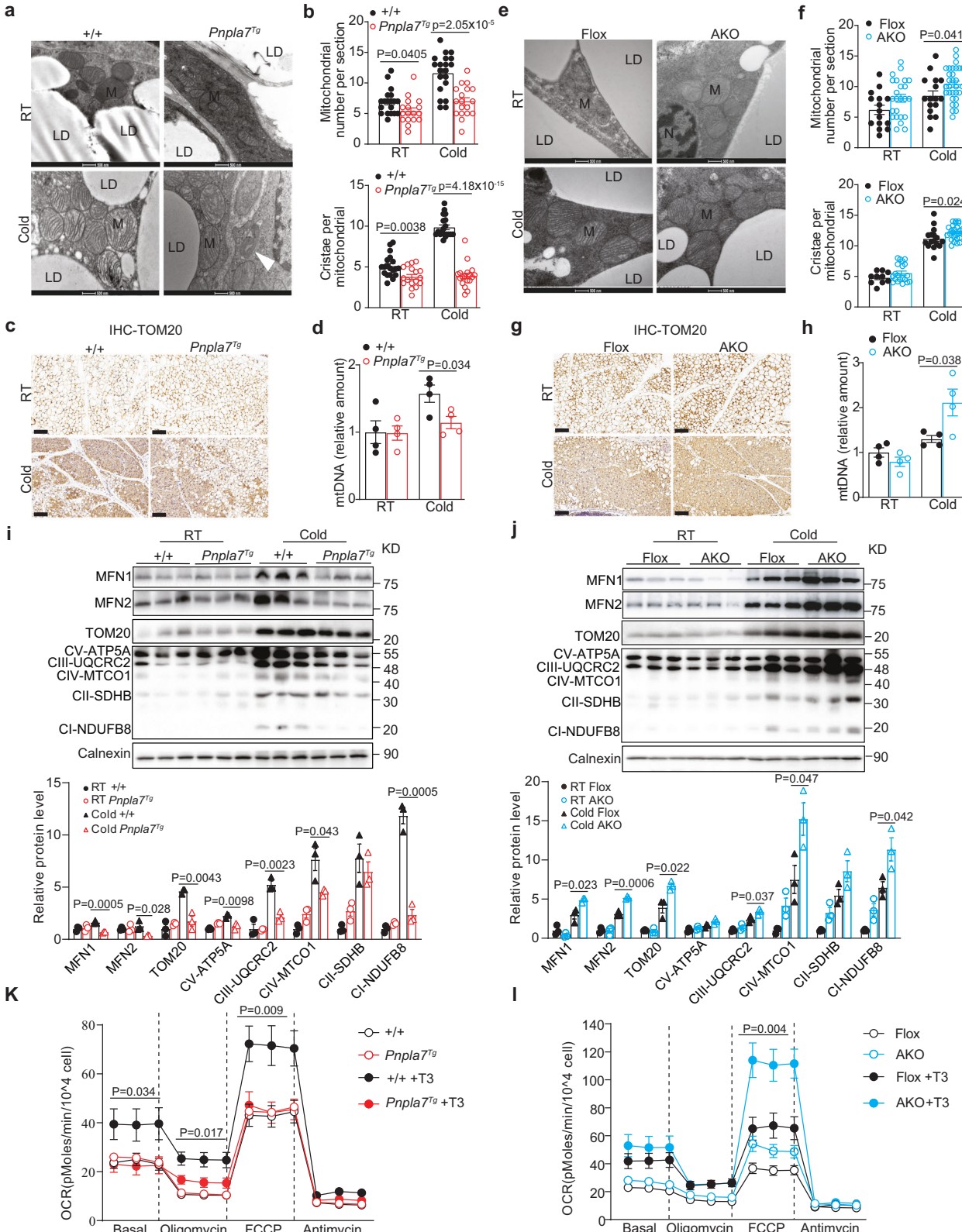

### PNPLA7 induces PINK1/Parkin-mediated mitophagy via Parkin mitochondrial recruitment

As PINK1/Parkin-mediated mitophagy is critical for mitochondrial degradation in the beige adipose tissue[25,46], we further evaluated whether PNPLA7-associated mitochondrial degradation is mediated through PINK1/Parkin-dependent mitophagy. Here, in the presence of CCCP or Valinomycin, PINK1 protein expression was induced in

adipocytes regardless of PNPLA7 protein expression (Fig. 5a and Supplementary Fig. 8a). On the other hand, Parkin expression was reduced significantly in PNPLA7 overexpressed adipocytes. The degree of Parkin protein reduction was much more obvious compared to the control adipocytes (Fig. 5a). Concomitantly, minor reduction in Parkin was observed in PNPLA7 depleted adipocytes when treated with CCCP or valinomycin (Supplementary Fig. 8a). Consistently, in the iWAT of

**Fig. 3 | PNPLA7 decreases mitochondrial content and mitochondrial oxidative respiration in the adipose tissue. a, b** Representative TEM images showing mitochondria (**a**, $n = 3$ biological replicates, scale bar = 500 nm) as well as quantification of mitochondrial number per section and cristae number per mitochondrion (**b**, RT: +/+ group: $n = 19$; $Pnpla7^{Tg}$ group: $n = 17$; Cold: +/+ group: $n = 20$; $Pnpla7^{Tg}$ group: $n = 18$) of iWAT from control and $Pnpla7^{Tg}$ male mice. The white arrow highlights the mitophagosome. Cristae number normalized by each section. Data are presented as mean ± SEM. (Two-tailed Student's $t$ test for 2-group comparisons). **c, d** Representative TOM20 immunohistochemical staining images (**c**, $n = 3$ biological replicates, scale bar = 100 μm) and relative mtDNA content (**b**, $n = 4$/group) of iWAT from control and $Pnpla7^{Tg}$ male mice. Data are presented as mean ± SEM. (Two-tailed Student's $t$ test for 2-group comparisons). **e, f** Representative TEM images showing mitochondria (**e**, $n = 3$ biological replicates, scale bar = 500 nm) as well as quantification of mitochondrial number per section and cristae number per mitochondrion (RT: Flox group: $n = 15$; AKO group: $n = 24$; Cold: Flox group: $n = 16$; AKO group: $n = 30$) of iWAT of Flox and $Pnpla7^{AKO}$ male mice.

Cristae number normalized by each section. Data are presented as mean ± SEM. (Two-tailed Student's $t$ test for 2-group comparisons). **g, h** Representative TOM20 immunohistochemical staining images (**c**, $n = 3$ biological replicates, scale bar = 100 μm) and relative mtDNA content (**b**, $n = 4$/group) of the iWAT from Flox and $Pnpla7^{AKO}$ male mice. Data are presented as mean ± SEM. (Two-tailed Student's $t$ test for 2-group comparisons). **i, j** Representative Immunoblot results and densitometry analysis of indicated protein levels in iWAT of control and $Pnpla7^{Tg}$ (**i**) as well as Flox and $Pnpla7^{AKO}$ (**j**) male mice ($n = 3$/group). Data are presented as mean ± SEM. (Two-tailed Student's $t$ test for 2-group comparisons). **k, l** Mitochondrial oxygen consumption rate of differentiated mature white and beige adipocytes from control and $Pnpla7^{Tg}$ (**k**) as well as control and $Pnpla7^{AKO}$ (**l**) male mice ($n = 3$ biological replicates). Data are presented as mean ± SEM. (Two-tailed Student's $t$ test for 2-group comparisons). The stromal vascular fraction (SVF) was isolated from iWAT of 3-week-old male mice and differentiated into mature adipocytes ex vivo. Source data are provided as a Source Data file.

cold-exposed mice, overexpression or ablation of PNPLA7 did not affect the expression of PINK1 in total lysate. In contrast, Parkin was reduced in iWAT of $Pnpla7^{Tg}$ mice, whereas cold exposure has limited effect on Parkin protein levels in iWAT of $Pnpla7^{AKO}$ mice (Fig. 5b and Supplementary Fig. 8b).

Parkin mitochondrial recruitment is one of the most important steps for PINK1/Parkin mediated mitophagy[25,36]. Thus, we next evaluated mitochondrial recruitment of PINK1 and Parkin in iWAT of $Pnpla7^{Tg}$ and $Pnpla7^{AKO}$ mice under cold exposure. Consistently, overexpression (Fig. 5c) or depletion (Supplementary Fig. 8c) of PNPLA7 did not affect PINK1 mitochondrial enrichment in iWAT of mice under RT. Upon cold exposure, PINK1 mitochondrial localization in both iWAT was reduced. On the other hand, unlike the wild type iWAT, abundant Parkin protein was detected in mitochondrial enriched fraction and less in cytosolic fraction of iWAT of $Pnpla7^{Tg}$ mice after cold exposure (Fig. 5c). Inversely, Parkin distribution was less in the mitochondrial fraction of iWAT from $Pnpla7^{AKO}$ mice under the same condition (Supplementary Fig. 8c). Concomitantly, LC3, an autophagy marker, exhibited a similar expression pattern as Parkin following cold exposure, whereas no significant difference was observed for P62 expression (Fig. 5c and Supplementary Fig. 8c). The enriched Parkin mitochondrial recruitment and LC3 and P62 proteins expression profiles were further confirmed in iWAT of $Ad$-$Pnpla7$ mice after cold exposure (Supplementary Fig. 8d, e). In addition, mitochondrial localization of EGFP-Parkin was stimulated in $Pnpla7^{Tg}$ adipocytes after incubation with 10 μm CCCP compared with that of control cells (Fig. 5d, e). In contrast, in PNPLA7 deficient cells, EGFP-Parkin colocalization with mitochondria was not observed (Supplementary Fig. 8f, g). These results suggest that PNPLA7 promotes mitochondrial recruitment of Parkin without altering PINK1 protein level both in vivo and in vitro.

To further evaluate if PNPLA7 regulation of mitochondrial degradation is a functional consequences of Parkin mitochondrial recruitment, we attempted to examine the phosphorylation status of poly-ubiquitin in the differentiated adipocytes derived from iWAT after CCCP or Valinomycin treatment. In addition, we also investigated the ubiquitination of MFN2 and TOM20, the two classical substrates of Parkin. Indeed, overexpression of PNPLA7 significantly increased phosphorylation of Poly-Ubiquitin at Ser65 as well as polyubiquitination of MFN2 and TOM20 (Fig. 5f). In contrast, depletion of PNPLA7 in adipocytes significantly attenuated the phosphorylation of Poly-Ubiquitin$^{Ser65}$, as well as poly-ubiquitination of Parkin's substrates, MFN2 and TOM20 under the same condition (Fig. 5g). More importantly, overexpression of full-length PNPLA7 proteins or PNPLA7$^{S983A}$ mutant proteins in PNPLA7 deficient adipocytes restored the phosphorylation of Poly-Ubiquitin$^{Ser65}$. Similarly, ubiquitination of two Parkin substrates, MFN2 and TOM20 were also restored in the presence of CCCP or valinomycin (Supplementary Fig. 8h, i). These results suggest

that PNPLA7-promoted mitochondrial degradation is dependent on Parkin mitochondrial recruitment and E3 ligase activity.

To assess whether the inhibitory effect of PNPLA7 on mitophagy is Parkin-dependent, we introduced AAV-Adipo-shParkin into iWAT of $Pnpla7^{Tg}$ and littermate control mice by orthotopic injection (Fig. 5h) as previously described[47]. Knocking down Parkin normalized the weight gained as well as the decreased mtDNA level in iWAT of $Pnpla7^{Tg}$ to that of the littermate control mice after prolonged cold exposure (Fig. 5i, j). Meanwhile, UCP1$^+$ and TOM20$^+$ beige adipocytes in iWAT of $Pnpla7^{Tg}$ mice were also restored (Fig. 5k). In addition, the lowered mitochondrial proteins such as MFN1, MFN2, TOM20, UCP1, and OXPHOS in the iWAT of $Pnpla7^{Tg}$ mice after prolonged cold exposure also resumed to the littermate control levels with the knockdown of Parkin (Fig. 5l). These results indicate that the inhibitory effect of PNPLA7 on iWAT browning is Parkin-dependent.

To further confirm the importance of mitochondrial-Parkin in the PNPLA7 inhibition of iWAT browning, we overexpressed a dominant negative mutant Parkin$^{R274W}$, which disrupts mitochondrial recruitment of Parkin to reduce mitophagy[36,48], in iWAT of $Pnpla7^{Tg}$ mice by orthotopic injection. Consistently, overexpression of Parkin$^{R274W}$ recapitulated the effect of knocking down Parkin, whereby the downregulated iWAT browning was restored in $Pnpla7^{Tg}$ mice after prolonged cold exposure (Supplementary Fig. 9a–e).

Taken together, these results suggest that PNPLA7 activates PINK1/Parkin-mediated mitophagy by promoting Parkin's mitochondrial recruitment to enhance mitochondrial degradation, thereby impairing iWAT browning.

### PNPLA7 interacts with Parkin to facilitate Parkin mitochondrial recruitment

To provide further insight into how PNPLA7 regulates Parkin mitochondrial recruitment, we determined the subcellular localization of PNPLA7 in the 3T3-L1 pre-adipocyte. As previously reported[42,43], a significant amount of PNPLA7 was found in the ER fraction (Fig. 6a). Interestingly, PNPLA7 also cofractionated with FACL4, a mitochondria-associated ER membrane (MAM)-specific marker, indicating that PNPLA7 proteins are localized to both the ER and MAM (Fig. 6a).

The subcellular localization of PNPLA7 on ER and MAM was further confirmed by immunostaining in 3T3-L1 adipocytes. Consistent with the results from cell fractionation, EGFP-PNPLA7 was found to co-localize with the ER-Tracker (Fig. 6b). In addition, colocalization of mCherry-PNPLA7 with the MAM marker FACL4 proteins was also observed (Fig. 6c). Moreover, the EGFP-PNPLA7 signals did not co-localize with the Mito-Tracker in adipocytes (Fig. 6d). These data demonstrate that PNPLA7 is predominantly located in the ER and MAM, where PINK1/Parkin-dependent mitophagy is initiated[49,50].

To further elucidate the molecular mechanisms underlying the regulatory function of PNPLA7 on Parkin mitochondrial recruitment,

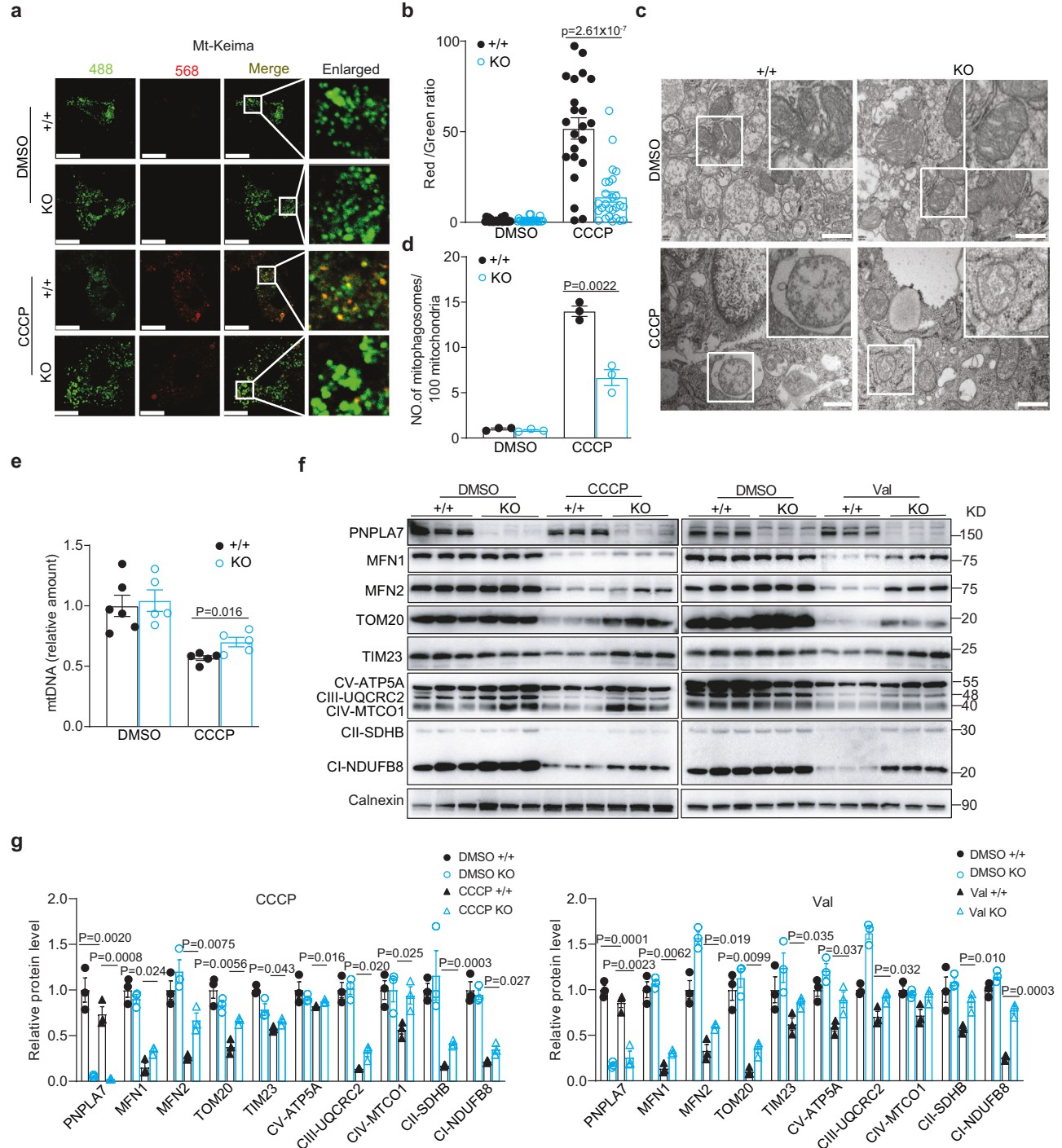

**Fig. 4 | Ablation of PNPLA7 decreases mitophagy in WAT adipocytes.** Stromal vascular fraction (SVF) was isolated from iWAT of 3-week-old control (+/+) or PNPLA7 knockout (KO) male mice and differentiated into mature adipocytes ex vitro. **a** Representative fluorescence images of cells stained with mito-Keima in differentiated adipocytes. Differentiated adipocytes were infected with mito-Keima lentivirus and then treated with DMSO or CCCP (10 μM) for 12 h (*n* = 3 independent experiments). The fluorescence image obtained by confocal microscopy after excitation at 488-nm and 568-nm are shown in green and red in the same cell, respectively. Scale bar = 20 μm. **b** Quantification of the relative ratio of fluorescence intensity (568 nm:488 nm) of the cells described in (**a**) (DMSO group: WT: *n* = 24; KO: *n* = 23; CCCP group: WT: *n* = 22; KO: *n* = 26). Data are presented as mean ± SEM. (Two-tailed Student's *t* test for 2-group comparisons). **c** Representative TEM images showing mitophagosome in differentiated mature adipocytes treated with DMSO or CCCP (10 μM) for 12 h (*n* = 3 independent

experiments). The white box highlights the mitochondria and mitophagosome. Scale bar = 500 nm. **d** Quantification of mitophagosome number per 100 mitochondria in the cells described in (**c**). (*n* = 3 independent experiments). Data are presented as mean ± SEM. (Two-tailed Student's *t* test for 2-group comparisons). **e** Relative mtDNA content in differentiated mature adipocytes treated with DMSO or CCCP (10 μM) for 24 h (DMSO: WT group: *n* = 6; KO group: *n* = 5; CCCP: *n* = 5/group). Data are presented as mean ± SEM. (Two-tailed Student's *t* test for 2-group comparisons). **f, g** Representative Immunoblot results (**f**) and densitometry analysis (**g**) of mitochondrial membrane, and OXPHOS protein levels in differentiated mature adipocytes treated with CCCP (10 μM) or Valinomycin (1 μM) for 24 h, respectively (*n* = 3 biological replicates). Data are presented as mean ± SEM. (Two-tailed Student's *t* test for 2-group comparisons). Source data are provided as a Source Data file.

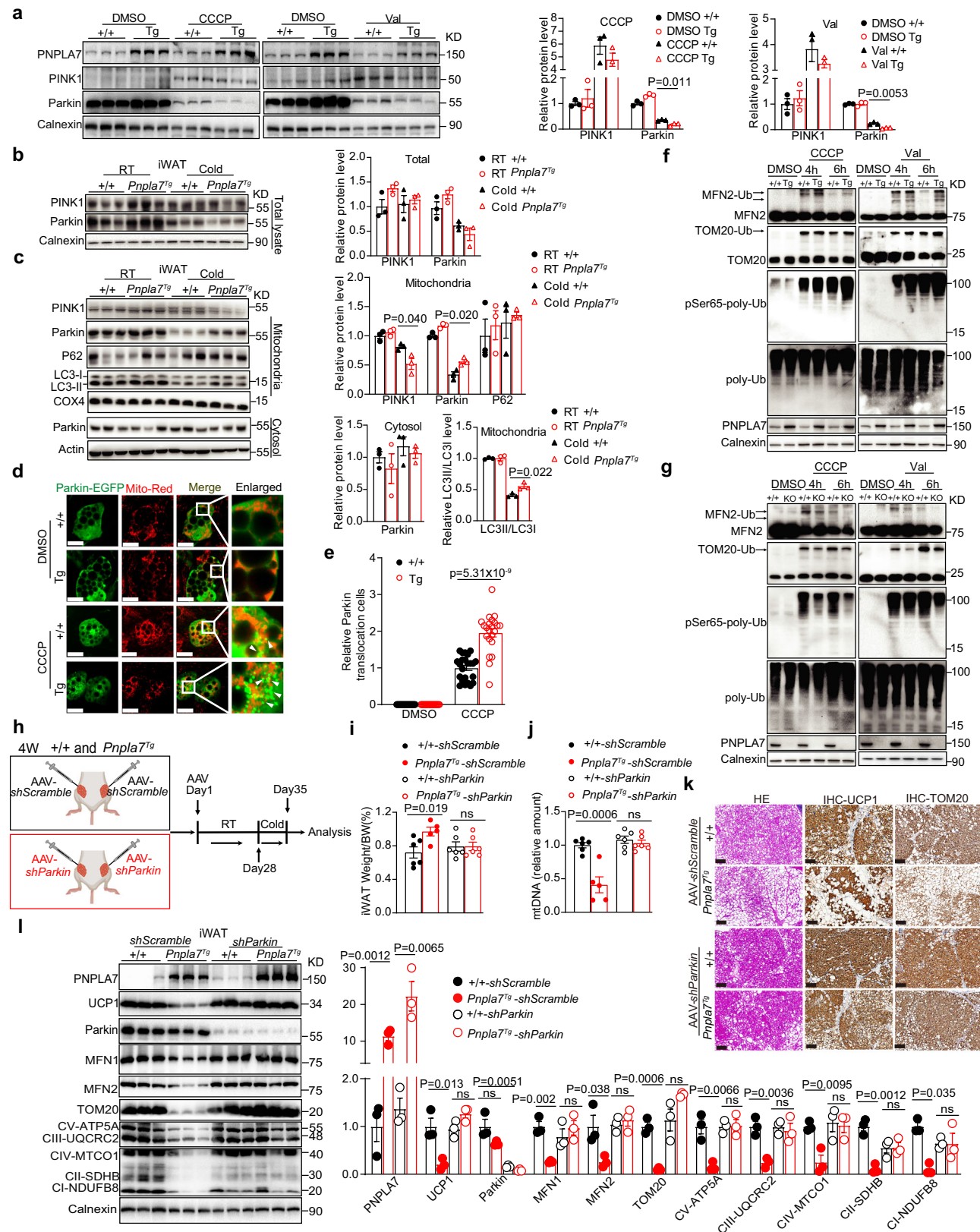

we examined the interaction between PNPLA7 and Parkin in iWAT of mice. Immunoprecipitation and Western blotting results demonstrated a strong interaction between endogenous Parkin and Flag-PNPLA7 in the iWAT of *Pnpla7*^Tg mice (Fig. 6e, f), suggesting that PNPLA7 forms a specific complex with Parkin in vivo. To map the interaction domains, a series of truncated PNPLA7 mutants were

generated (Fig. 6g). Both full-length PNPLA7 (aa1-1352) and truncations that contain C-terminal Patatin domain (aa36-1352 and aa942-1352) interacted with Parkin. This interaction was abolished in PNPLA7 (aa1-941) (Fig. 6h, i). The data indicate that the C-terminal region of PNPLA7, which consists of the Patatin domain, is required for its interaction with Parkin. Next, we introduced PNPLA7 full-length and truncated

**Fig. 5 | PNPLA7 enhances Parkin mitochondrial recruitment and E3 ligase activity. a**–**c** Representative Immunoblot results and densitometry analysis of the indicated proteins in differentiated mature inguinal adipocytes (**a**), total lysate (**b**), mitochondrial and cytosolic fraction (**c**) of iWAT isolated from control and *Pnpla7*[Tg] mice (*n* = 3 biological replicates). Data are presented as mean ± SEM. (Two-tailed Student's *t* test for 2-group comparisons). **d**, **e** Colocalization of Parkin with mitochondria (**d**) and quantification of the mitochondrial EGFP-Parkin positive cells (**e**) in differentiated adipocytes with or without PNPLA7 overexpression. Mitochondria were stained with Mito-tracker (red). EGFP-Parkin translocation to mitochondria was analyzed using confocal microscopy. (*n* = 3 biological replicates, scale bar = 20 μm). Data are presented as mean ± SEM. (Two-tailed Student's *t*-test for 2-group comparisons). **f**, **g** Representative Immunoblot results of the indicated proteins in differentiated mature adipocytes obtained from control and *Pnpla7*[Tg] (**f**) as well as control and PNPLA7 knockout (**g**) male mice. (*n* = 3 biological replicates). **h** Illustration of the Adeno-Associated Virus (AAV)-mediated knocking down strategy of Parkin in iWAT of *Pnpla7*[Tg] *mice* and littermate control. 4-week-old control (+/+) and *Pnpla7*[Tg] (Tg) mice were injected with recombinant adeno-

associated virus of AAV-Adipo-*shscramble* or AAV-Adipo-*shParkin* into the iWAT. After 28 days injection, the mice were exposed to 6 °C for additional 7 days. These mice were fed with normal chow diet. The mouse picture was created in BioRender. Ji, X. (2025) https://BioRender.com/v1sk48t. **i**, **j** The iWAT weight ratio (**i**) and relative mtDNA content (**j**) of control (+/+) and *Pnpla7*[Tg] (Tg) mice with or without *Parkin* knockdown as described in (**h**). (*shScramble*: +/+ group: *n* = 6; *Pnpla7*[Tg] group: *n* = 5; *shParkin*: *n* = 6/group). Data are presented as mean ± SEM. (Two-tailed Student's *t* test for 2-group comparisons; ns indicates no statistical significance). **k** Representative H&E, UCP1, and TOM20 immunohistochemical staining images of iWAT sections from control (+/+) and *Pnpla7*[Tg] (Tg) mice with or without *Parkin* knockdown as described in (**h**). (*n* = 3/group). Scale bar = 100 μm. **l** Representative Immunoblot results and densitometry analysis of the indicated proteins in iWAT control (+/+) and *Pnpla7*[Tg] (Tg) mice with or without *Parkin* knockdown as described in (**h**). (*n* = 3/group). Data are presented as mean ± SEM. (Two-tailed Student's *t* test for 2-group comparisons; ns indicates no statistical significance). Source data are provided as a Source Data file.

mutants into PNPLA7-deficient adipocytes to explore the biological significance of PNPLA7-Parkin interaction in the regulation of Parkin mitochondrial recruitment. As shown in Fig. 6j, in PNPLA7-deficient adipocytes treated with CCCP, the EGFP-Parkin fluorescence signal was found to be diffusely distributed in the cytoplasm. Interestingly, when full-length (aa1-1352) or truncated PNPLA7 (aa942-1352) were overexpressed, EGFP-Parkin proteins aggregated and formed punctate structures. This was not observed when the N-terminal half of PNPLA7 (aa1-941) was overexpressed (Fig. 6j). These findings indicate that the C-terminal region of PNPLA7, which consists of the Patatin domain, is required for the interaction between PNPLA7 and Parkin to facilitate Parkin mitochondrial recruitment.

### PNPLA7 disrupts PKA-mediated phosphorylation of Parkin

Considering the inhibitory role of PNPLA7 in iWAT browning through the promotion of Parkin-mediated mitophagy, we asked if the interaction between PNPLA7 and Parkin is modulated by β3-AR signaling. Indeed, immunoprecipitation assay showed that the interaction between PNPLA7 and Parkin was significantly reduced in both iWAT of WT mice (Fig. 7a) and in vitro differentiated adipocytes (Fig. 7b) when treated with CL 316,243. These findings are consistent with the downregulation of PNPLA7 in BAT and iWAT upon the activation of β3-AR signaling (Fig. 1c, e), suggesting a negative role for PNPLA7 in β3-AR signaling.

In β3-AR signaling, PKA serves as the primary downstream effector, whereby its activation significantly enhances Parkin phosphorylation. This, in turn, reduces Parkin mitochondrial recruitment and thus prevents PINK1/Parkin-mediated mitophagy[26,39]. We further evaluated the impact of PNPLA7 on PKA-regulated mitochondrial recruitment of Parkin in adipocytes. CCCP treatment promoted the mitochondrial recruitment of Parkin in adipocytes as evidenced by the colocalization analysis between EGFP-Parkin and mitochondria. However, such colocalization was abolished when cells were co-treated with CL 316,243 (Fig. 7c). Intriguingly, the loss of EGFP-Parkin-mitochondria colocalization was re-established in PNPLA7 overexpressing adipocytes, but not in PNPLA7 deficient adipocytes (Fig. 7c). These data suggest that PNPLA7 counteracts the inhibitory function exerted by PKA signaling pathway on Parkin mitochondrial recruitment.

Subsequently, we conducted a co-immunoprecipitation assay using p-PKA substrate antibody to investigate the impact of PNPLA7 on PKA-mediated phosphorylation of Parkin in iWAT following CL 316,243 injection into the mice. Our findings revealed that p-PKA substrate antibody was able to pull down low levels of phosphorylated Parkin in the iWAT of wild type mice. Upon CL 316,243 treatment, the amount of immunoprecipitated phospho-Parkin increased significantly. However, this robust immunoprecipitation of phospho-Parkin was

attenuated in the iWAT of *Pnpla7*[Tg] mice (Fig. 7d). Conversely, compared to *flox/flox* control mice, CL 316,243 injection enhanced phosphorylation of Parkin in iWAT of *Pnpla7*[AKO] mice (Fig. 7e). Consistent with our in vivo data, PNPLA7-deficient adipocytes treated with CL 316,243 displayed augmented Parkin phosphorylation (Fig. 7f) which was alleviated by the pretreatment with PKA-specific inhibitor H89 (Fig. 7f). These results together support the notion that PNPLA7 impedes browning of iWAT by inhibiting PKA-mediated phosphorylation of Parkin, thereby facilitating Parkin mitochondrial recruitment and mitophagy for mitochondrial degradation (Fig. 7g).

## Discussion

Increasing evidence suggests that promoting the development of beige adipocytes in WAT through browning is crucial for adaptive thermogenesis in response to external stimuli, thereby potentially improve metabolic health[10,51,52]. PINK1/Parkin mediated ubiquitin-dependent mitophagy, a selective autophagy for mitochondrial clearance, is critical for the maintenance of healthy mitochondrial network in adipose tissue. However, the precise regulatory mechanism underlying PINK1/Parkin-mediated mitophagy in iWAT remains largely unknown. In this study, we identified PNPLA7 as a negative regulator of iWAT browning in mice. Its expression is tightly controlled by ambient temperature and thermogenic signals. Gain-and loss-of function studies revealed that PNPLA7 inhibits iWAT browning by promoting PINK1/Parkin-mediated mitophagy. Mechanically, we found that PNPLA7 interacts with Parkin to block PKA-mediated phosphorylation of Parkin under cold environments, thereby increasing mitochondrial recruitment of Parkin and promoting mitophagy for mitochondrial degradation, resulting in the inhibition of iWAT browning. Taken together, these findings provide a direct mechanistic link between PNPLA7 and PINK1/Parkin-mediated mitophagy and browning of iWAT.

Previous studies have shown that PNPLA7 is highly expressed in liver, skeletal muscle, adipose tissue, and reduced by insulin in 3T3-L1 cells[40]. Although PNPLA7 is enriched in BAT and iWAT, it is downregulated by cold stress or β3-AR agonist treatment both in vivo and in vitro. The downregulated PNPLA7 protein expression is inversely correlated to the induction of thermogenic protein UCP1. These findings indicated that PNPLA7 is negatively regulated by thermal stress or adrenergic signal, therefore plays an important role in adaptive thermogenesis. Corroboratively, PNPLA7 null mice exhibits reduced adiposity index with browning phenotypes observed in WAT while BAT remains unaffected[44]. However, adipose stromal cells (ASCs) derived from PNPLA7 null mice display normal LD accumulation during differentiation without browning indication[44]. In contrast, ablation of PNPLA7 gave rise to enhanced mitochondria respiration in our in vitro differentiated SVF isolated from iWAT of our animal models using Triiodothyronine (T3), a potent hormone in differentiating brown

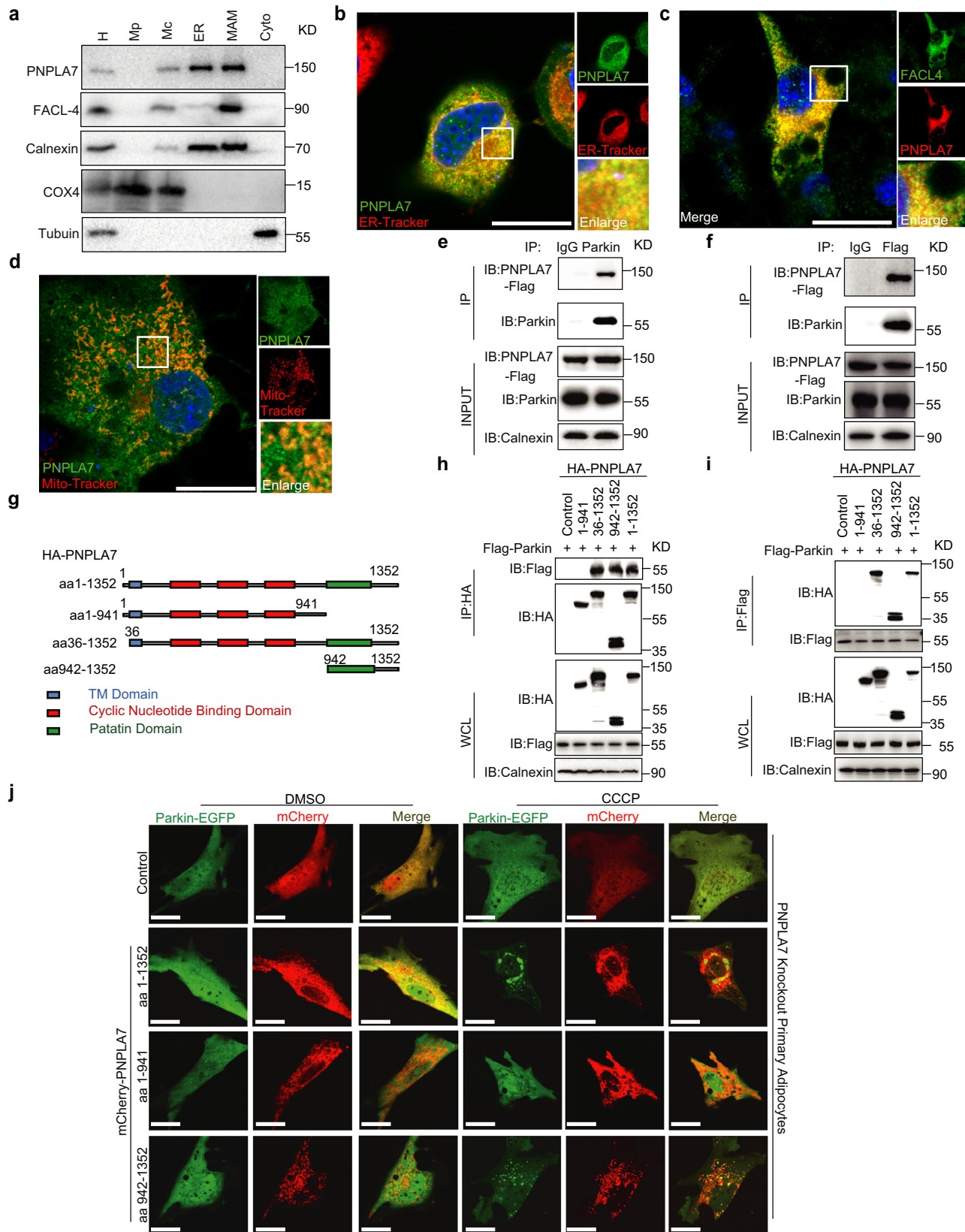

adipocytes[45]. In parallel, overexpression of PNPLA7 repressed mitochondrial respiration. In addition, our in vitro experiments also show that primary adipocytes differentiated from SVF cells of *Pnpla7^AKO* presented less mitophagy when induced by CCCP or Valinomycin, whereas *Pnpla7^Tg* primary adipocytes displayed enhanced mitophagy under the same treatment. Collectively, our data using complementary in vitro adipocyte culture models demonstrate a cell autonomous regulation of browning and mitophagy by PNPLA7. Of note, PNPLA7 does not affect basal adipocyte differentiation as consistently demonstrated in the current study and a previous report[44]. Further work is warranted to understand why PNPLA7 is dispensable in regular adipocyte differentiation.

**Fig. 6 | PNPLA7 is a MAM protein and interacts with Parkin to facilitate mitochondrial translocation of Parkin. a** Representative Immunoblot results of PNPLA7 protein levels in the indicated organelle fractions isolated from 3T3-L1 cells. H: homogenate, Mp: pure mitochondrial fraction, Mc: crude mitochondrial fraction, ER: the endoplasmic reticulum, MAM: mitochondria-associated membrane, Cyto: cytosol. Organelle fractions were verified by testing the presence of specific markers: Calnexin for ER and MAMs, FACL-4 for MAMs, COX4 for mitochondrial, tubulin for cytosol. Representative results from 3 independent experiments were shown. **b** Representative immunofluorescence image of EGFP-PNPLA7 colocalization with ER in 3T3-L1 cells. EGFP-PNPLA7 localization to ER was analyzed by confocal microscopy. Scale bar = 20 μm. (*n* = 3 biological replicates). **c** Representative immunofluorescence image of mCherry-PNPLA7 colocalization with mitochondria-associated membrane in 3T3-L1 cells. Mitochondria-associated membranes were stained for FACL-4 (green), mCherry-PNPLA7 localization to mitochondria-associated membrane was analyzed by confocal microscopy. Scale bar = 20 μm. (*n* = 3 biological replicates). **d** Representative immunofluorescence image of EGFP-PNPLA7 colocalization with mitochondria in 3T3-L1 cells. EGFP-PNPLA7 localization to mitochondria was analyzed by confocal microscopy. Scale bar = 20 μm. (*n* = 3 biological replicates). **e, f** Representative co-immunoprecipitation Immunoblot data of endogenous Parkin with PNPLA7 in iWAT. *Pnpla7*^Tg^ mice iWAT total lysates were immunoprecipitated with anti-Parkin (**e**) or anti-Flag (**f**) antibodies, respectively. Calnexin was used as a protein loading control. IgG, immunoglobulin G. (*n* = 3 biological replicates). **g** Schematic diagram of construction of PNPLA7 truncations. Full length of PNPLA7(1-1352aa), N-terminal region (1-941aa), Transmembrane domain (36-1352aa), and C-terminal region (942-1352aa). All truncations carry HA-tag at the carboxyl terminus. **h, i** Representative co-immunoprecipitation Immunoblot data of Flag-Parkin with different truncations of HA-PNPLA7 in HEK293T cells. HEK293T cells were transfected with Flag-Parkin and different truncations of HA-PNPLA7 for 48 h. Cell total lysates were immunoprecipitated with anti-HA (**h**) or anti-Flag (**i**) antibodies, respectively. HA-vector was used as a negative control of immunoprecipitation. Calnexin was used as a protein loading control. (*n* = 3 biological replicates). **j** Mitochondrial translocation of Parkin in PNPLA7-depleted mature adipocytes overexpressing different PNPLA7 truncations. SVF from iWAT of PNPLA7 KO mice was differentiated into adipocytes ex vitro. Immunofluorescence images of EGFP-Parkin (green) and mCherry-PNPLA7 (red) were analyzed by confocal microscopy. Scale bar = 20 μm. (*n* = 3 biological replicates). Source data are provided as a Source Data file.

The upregulation of UCP-1 is a determinant for browning. In addition, transcriptional factors such as PPARγ, PRDM16, and PGC-1α are also induced[19,53,54]. Nevertheless, effective browning requires the maintenance of mitochondrial homeostasis, which encompasses mitochondrial biogenesis, mitochondrial fusion-fission dynamics, and mitophagy-mediated recycling[18,27]. In this study, since the mRNA levels of genes involved in mitochondrial biogenesis under the basal condition remained unchanged in *Pnpla7*^AKO^, *Pnpla7*^Tg^, or *Ad-Pnpla7* mice, we focused our investigation on mitochondrial homeostasis during thermogenesis. Under cold exposure, UCP1 induction and mitochondria number and morphology are inversely related to PNPLA7 expression in iWAT. In addition, CCCP treatment on PNPLA7-deficient adipocytes resulted in a reduction of mitophagosomes. Our observations are consistent with recent reports that genetic inactivation of serine/threonine-protein kinase 3 (STK3) and STK4 decreases adipocyte mitophagy, leading to the promotion of mitochondrial mass and function and UCP1 stabilization in beige adipose tissue[29]. Similarly, inhibiting mitophagy by specific deletion of Atg5 or Atg12 in UCP1⁺-adipocyte prevents beige adipocyte loss after the withdrawal of external stimuli, thereby maintaining high thermogenic capacity[27,55]. Notably, UCP1 protein levels and mitochondria morphology in BAT are not affected by PNPLA7 expression, suggesting that PNPLA7-mediated mitophagy is not directly involved in the regulation of mitochondria homeostasis in BAT. The phenotypic differences between iWAT and BAT may arise due to the fundamental difference in mitophagy in WAT during browning, exhibiting cellular specificity[23,27,30,37]. The role of PNPLA7 in BAT remains to be investigated.

PINK1, the serine/threonine kinase, and Parkin, the E3 ubiquitin ligase, are two key molecules mediating mitophagy and play important roles in adipose tissue remodeling[27,37,38]. Previously, it was shown that deletion of Parkin in WAT rather than BAT promotes to the maintenance of browning and ameliorates HFD feeding or aging-induced obesity and glucose disorder[24,30]. It was suggested that the absence of Parkin inhibits mitophagy and promotes biogenesis through Nqo1-Pgc-1α regulation[30]. Here, we report that PNPLA7 regulates Parkin-mediated mitochondria homeostasis. The first indication of Parkin involved in PNPLA7-associated mitophagy is evident when Parkin protein strongly decreases in response to mitochondrial damaging agent (CCCP & Valinomycin) treatment in PNPLA7 overexpressing adipocytes when compared to wild type or PNPLA7 deficient adipocytes. Further investigation with cold exposure showed the presence of enhanced Parkin proteins in the mitochondrial fraction of iWAT of *Pnpla7*^Tg^ and *Ad-Pnpla7* mice. In addition, overexpression of PNPLA7 induced Parkin E3 ligase activity as evidenced by the increase in polyubiquitination of Parkin substrates. Finally, Parkin activity inhibition or knocking down Parkin in *Pnpla7*^Tg^ mice rescues the impaired browning capacity. Taken together, PNPLA7 role in anti-browning in iWAT is associated with Parkin-mediated mitophagy.

There are three major factors regulating Parkin E3 ligase activity. These include Parkin binding to pSer65-Ub, Parkin mitochondrial recruitment, and Parkin phosphorylation status[56–60]. According to the current prevailing hypothesis, pSer65-Ub serves as a potent receptor for recruiting Parkin to mitochondria. The binding of pSer65-Ub to Parkin changes Parkin conformation and activates its E3 ligase activity[61,62]. Importantly, overexpression of PNPLA7 increases pSer65-Ub level after CCCP or Valinomycin treatment. Since the presence of PNPLA7 did not affect PINK1 activity, it is highly plausible that the increased pSer65-Ub levels is due to PNPLA7 promoting the recruitment of pSer65-Ub to Parkin after Parkin undergoes mitochondrial recruitment. This, in turn, induces Parkin phosphorylation and further results in a feedback response with an increase in pSer65-Ub levels. Previously, PNPLA7 has been reported to be a lysophospholipase that hydrolyzes *sn*-1 ester of LPC preferentially in vitro and controls hepatic choline and methionine metabolism[42,44]. Nevertheless, our overexpression and knockout studies indicate that PNPLA7 has no effect on LPC/PC and LPE/PE content in iWAT under both RT and cold environment. Consistently, others have shown that PNPLA7 null-mice do not display significant change in LPC/PC content in the BAT[44]. Collectively, the data indicate that PNPLA7 has low lysophospholipase activity in the adipose tissue, however, PNPLA7 protein per se has regulatory function. In addition, overexpression of PNPLA7 or PNPLA7 (S983A) significantly increases the ubiquitination of MFN2, TOM20, and pSer65-Ub in PNPLA7-deficient adipocytes when treated with CCCP or Valinomycin, suggesting that PNPLA7's role in regulating mitophagy in beige adipocytes is independent of its lysophospholipase activity.

In establishing the function of PNPLA7 in the regulation of mitophagy, we provided convincing evidence demonstrating that PNPLA7 is also enriched in MAMs besides its classic localization in the ER[42,43]. MAMs are regions of close physical connection between the mitochondrial outer membrane and the ER membranes. These are the sites for mitochondrial DNA synthesis and fission, calcium exchange, and lipid biosynthesis, and are indispensable for mitochondrial dynamics and function[63–65]. Increasing evidence has shown that the core proteins involved in PINK1/Parkin-mediated mitophagy are also located in the MAMs and are involved in the regulation of MAMs integrity and function[57,66]. In this study, we showed that PNPLA7 physically interacts with Parkin via its C-terminal region containing the Patatin domain and PNPLA7 has a drastic effect on Parkin mitochondrial recruitment. Thus, it is possible that MAM-localized PNPLA7 has a proximity advantage to

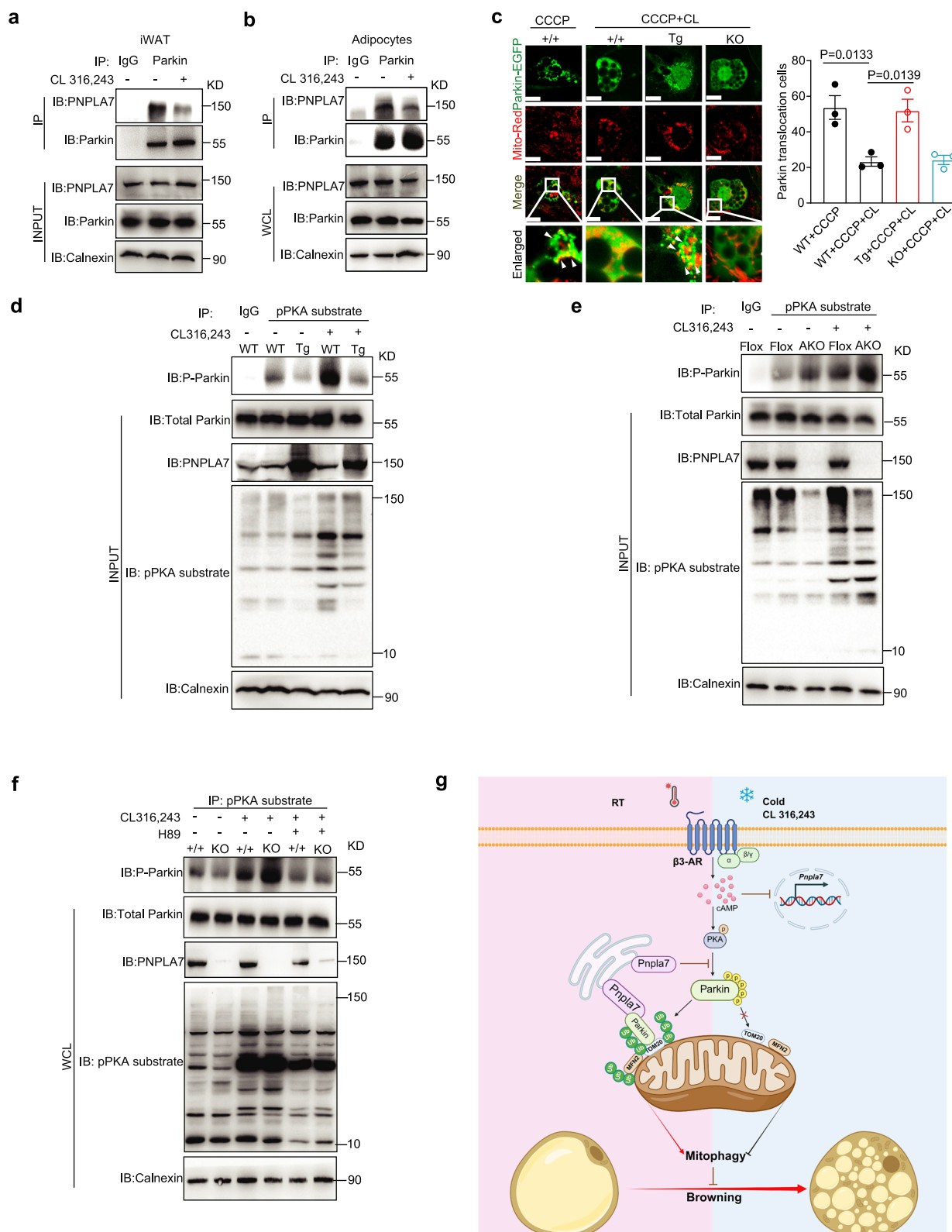

Parkin and, through this interaction, Parkin is able to translocate towards mitochondria upon external stimulation.

The third factor that determines Parkin E3 ligase activity is its phosphorylation status. Under chronic cold acclimation, activation of β3-AR-cAMP-PKA signaling pathway expands the oxidative capacity of WAT through converting it into a tissue resembling BAT. Inhibition of PINK1/Parkin-mediated mitophagy during the browning process is critical and can be achieved through PKA-mediated impairment of Parkin recruitment to mitochondria in beige cells[26,67]. The PKA phosphorylates Parkin, resulting in its inability to translocate to mitochondria[26]. This process is abolished when PNPLA7 is over-expressed in the iWAT of transgenic mice. Although our data support a role for PNPLA7 in regulating Parkin phosphorylation in iWAT, further experiments are required to elucidate the detailed molecular

**Fig. 7 | PNPLA7 promotes Parkin mitochondrial translocation by inhibiting PKA-mediated phosphorylation of Parkin. a, b** Representative co-immunoprecipitation Immunoblot data of endogenous Parkin with PNPLA7 in iWAT (**a**) and inguinal adipocytes (**b**) of wild-type mice. Calnexin was used as a protein loading control. IgG: immunoglobulin G. (*n* = 3 biological replicates). **c** Stromal vascular fraction (SVF) was isolated from iWAT of 3-week-old control (+/+) and *Pnpla7*^*Tg*^ (Tg) and PNPLA7 knockout (KO) male mice and differentiated into mature adipocytes ex vivo. Mitochondrial translocation of Parkin was analyzed in adipocytes with PNPLA7 overexpression or knockout. Immunofluorescence imaging of EGFP-Parkin (green) and Mito-Red (red) was analyzed by confocal microscopy and quantification of Parkin colocalization with the mitochondria analyzed by Image J. Scale bar = 20 μm. (*n* = 3 independent experiments) Data are presented as mean ± SEM. (Two-tailed Student's *t* test for 2-group comparisons). **d, e** Co-immunoprecipitation Immunoblot data of endogenous p-Parkin with p-PKA substrate in iWAT of control and *Pnpla7*^*Tg*^ (**d**) as well as control and *Pnpla7*^*AKO*^ (**e**) mice. 10-week-old control (+/+) and *Pnpla7*^*Tg*^ (Tg) mice were injected with saline or CL 316,243 (1 mg/kg) via tail vein for 30 min. Co-immunoprecipitated endogenous phosphorylated Parkin (P-Parkin) protein was detected by anti-Parkin antibody. The input protein level of Parkin, PNPLA7, p-PKA substrate, and calnexin were detected by the indicated antibodies, respectively. IgG immunoglobulin G. (*n* = 3 biological replicates). **f** Representative co-immunoprecipitation Immunoblot data of endogenous p-Parkin with p-PKA substrate in adipocytes of control and PNPLA7 KO mice. The SVF was isolated from iWAT of 3-week-old wild-type (+/+) and PNPLA7 KO mice and differentiated into adipocytes ex vitro. Then adipocytes were incubated with saline or CL 316,243 (1 μM) for 30 min, with or without pretreatment with H89 (1 μM) for 1 h. Co-immunoprecipitated endogenous phosphorylated Parkin (P-Parkin) protein was detected by anti-Parkin antibody. The input protein level of Parkin, PNPLA7, p-PKA substrate, and calnexin were detected by the indicated antibodies, respectively. (*n* = 3 biological replicates). **g** A schematic depicting the role of PNPLA7 in inhibiting browning of iWAT. This Illustration created in BioRender. Ji, X. (2025) https://BioRender.com/5c8g75r. Source data are provided as a Source Data file.

mechanism. Collectively, PNPLA7 regulates the phosphorylation of pSer65-Ub by disturbing Parkin mitochondrial recruitment and the phosphorylation of Parkin mediated by PKA.

In conclusion, in addition to regulate VLDL secretion, choline and methionine metabolism in the liver, PNPLA7 functions as a molecular chaperone in WAT, tethering Parkin to mitochondria via its C-terminal region containing the Patatin domain. This critical step, governed by PNPLA7 is a prerequisite for Parkin's subsequent conformational change and activation as E3 ligase in order to induce mitophagy. Thus, our data fill in a key missing piece of the mitophagy molecular puzzle by providing a feasible mechanism of Parkin recruitment in mitophagy regulation. In brief, PNPLA7 prevents premature activation of browning in WAT in physiological conditions and reverses browning in WAT after the withdrawal of thermogenic stimulation through the regulation of Park mitochondrial localization.

## Methods

### Animals

Adipose-specific PNPLA7 transgenic mice (*Pnpla7*^*Tg*^) were generated by subcloning mouse *Pnpla7* (NCBI Gene ID: 241274) into pl253-Ap2-neo vector containing the 7.9-kb aP2 gene promoter. Founder PNPLA7 transgenic mice were screened by PCR using primers that specifically detect the transgene but not endogenous *Pnpla7* (aP2-: F: GAG CAA CCA CAG TGC ATG CT, R: TGG GTA CCT GCG CTT GAT AGA). The aP2-PNPLA7 transgenic mice were maintained by crossing the heterozygous transgenic mice to wild-type C57BL/6J mice.

The CRISPR-Cas9 system was used to generate PNPLA7 whole body knockout mice, the single guide RNAs (sgRNA) were design as sgRNA1: AAC CTC ATG GCC CAC ATC ACT GG; sgRNA2: GTG ACG GTG TAG GAT CCC GTG GG. sgRNAs directed Cas9 endonuclease cleavage in intron 6–7 and intron 7–8, resulting in exon 7 deficiency by homologous recombination. And the injected zygotes were transplanted into the oviduct of pseudo-pregnant female mice to obtain germline transmission. Global PNPLA7 knockout (*Pnpla7*^*−/−*^) mice were generated by cross-breeding heterozygote with each other. Founder PNPLA7 knockout mice were screened by PCR using primers that specifically detect the Exon7 (F1: ACG CCC TCG GAT TCT TCT GTA; R1: AGG CCA CTG CCC CAC TAAC; F2: AGC ACG CCC TCG GAT TCT T, R2: GCC TGG GCT ACA TAG TGA GTT CT). A 407 bp product was detected in the wild type mice, while a 451 bp product was detected in the KO mice. Both fragments can be seen in the heterozygote.

To generate the *Pnpla7*^*flox/flox*^ mice, a targeting vector with 2.6 kb genomic fragment containing exons 7 was inserted between two *loxP* sites. An upstream 4963 bp fragment and a downstream 4957 bp fragment was used as homology arms. The targeting vectors were linearized and electroporated into ES cells and microinjected into the C57BL/6J blastocysts to obtained the *Pnpla7* floxed mice. Adipose tissue-specific PNPLA7 knockout (*Pnpla7*^*AKO*^) mice were generated by cross-breeding *Pnpla7*^*flox/flox*^ mice with *Adipoq*^*Cre*^ (Jackson Laboratory, stock #010803). Genotyping of mice harboring the *Pnpla7* flox allele was performed with F: TTG GAG AAG GCT TGT CAT TG; R: ATG AGG TTC AAT CCT CAG AG, resulting in a 366 bp of wild-type allele and 432 bp of flox allele. Genotyping of mice expressing Adiponectin-Cre was performed with primers Adiponectin-Cre-F: CGT ATA GCC GAA ATT GCC AG and Adiponectin-Cre-R CAA AAC AGG TAG TTA TTC GG resulting in a 200 bp product.

All of the mice were maintained in a 12-h light/dark cycle at 25 °C with free access to water and standard chow diet (LAD 0020) in the animal facility (specific-pathogen free) of Nanjing Medical University. Unless noted specifically, male animals were used for the experiments. Genotype, age, and number of mice for each experiment are indicated in the appropriate figure legends. All animal protocols were reviewed and approved by the Animal Ethics Committee of Nanjing Medical University and followed the National Institute of Health guidelines on the care and use of animals.

### Cold exposure and β₃-AR agonist CL 316,243 treatment

For chronic cold exposure[68,69], 8-week-old male mice were individually caged with minimal bedding and free access to food and water at 6 °C for 7 days. For the β₃-AR agonist, mice were injected with CL 316,243 daily at a dose of 1 mg/kg for 7 days.

### Indirect calorimetry

Oxygen consumption ($VO_2$), carbon dioxide production ($VCO_2$), and energy expenditure were measured by indirect calorimetry in an open-circuit Oxymax chamber, a component of the Comprehensive Lab Animal Monitoring System (CLAMS; TSE Phenomaster)[70,71]. The 8-week-old male mice were allowed to acclimatize in the metabolic chambers for 48 h before the starting the experiment to minimize stress form the housing change. $CO_2$ and $O_2$ levels were collected every 40 min for each mouse over a period of 96 h. For the CL316,243 treatment experiment, the mice were injected intraperitoneally with CL316,243 at a dose of 1 mg/kg body weight after 48 h of measurements during 96 h period. Upon completion of the experiment, the oxygen consumption and energy expenditure were calculated.

### Cell lines and culture

HEK293T, HEK293A, and 3T3-L1 cells were gift from Prof. Peng Li (Zhengzhou University). Cells were cultured in Dulbecco's modified Eagle's medium (DMEM) containing 10% fetal bovine serum (FBS) and 1% P/S. All the cells were cultured at 37 °C in an incubator with 5% $CO_2$ in the atmosphere.

## SVF cells and adipocytes isolation and differentiation

For primary iWAT stromal vascular fractions (SVFs) and mature adipocytes isolation[45]. iWAT from 3-week-old mice was digested for 60 min at 37 °C by collagenase I (Sigma-Aldrich, C5138) in 1× PBS with BSA (10 mg/ml). The digested cell suspension was centrifuged and the pelleted cells were resuspended, and filtered through a 100-μm cell strainer. Then the SVF cells were plated on 6-well plates in DMEM/F12 supplemented with 10% FBS and 1% P/S at 37 °C in a humidified 5% $CO_2$ incubator. Primary SVF cells were cultured in DMEM/F12 containing 10% FBS before inducing adipogenic differentiation. Beige adipocyte differentiation was induced in primary SVF cells by treating confluent cells for 96 h in medium containing 10% FBS, 5 mM dexamethasone, 0.5 μg/mL insulin, 0.5 mM isobutylmethylxanthine (IBMX), 1 μM rosiglitazone and 1 nM T3. Four days after induction, cells were switched to maintenance medium containing 0.5 μg/mL insulin, 1 nM T3, 1 μM rosiglitazone and 10% FBS for additional 2 days. White adipocyte differentiation was induced in primary SVF cells by treating confluent cells for 48 h in medium containing 10% FBS, 5 mM dexamethasone, 0.5 μg/mL insulin, 0.5 mM IBMX, 1 μM rosiglitazone. Two days after induction, cells were maintained in medium containing 0.5 μg /mL insulin, 1 μM rosiglitazone and 10% FBS for additional 4 days. All chemicals for cell culture were obtained from Sigma-Aldrich.

## Plasmid constructs and transfection

PNPLA7 (NM_146251.4) complementary DNAs (cDNAs) were generated by reverse transcription PCR from RNAs of iWAT and inserted into pcDNA3.3 vector with a C-terminal HA tag. Subcloning was performed to generate truncations of HA-PNPLA7 constructs. All the HA-PNPLA7 truncations were generated in pcDNA3.3 vector and mCherry-PNPLA7 was generated by inserting *Pnpla7* into pmCherry-N1 vector. PNPLA7$^{S983A}$ point mutation was generated using the PCR-based QuikChange® site-directed-mutagenesis method with Pfu polymerase (Stratagne) following the manufacturer's instructions[43]. The Flag-Parkin and GFP-Parkin were generated by inserting Parkin into pcDNA3.3 vector with a C-terminal Flag tag and pEGFP-N1 vector. All plasmid constructs were confirmed by DNA sequencing. Expression of all constructs was confirmed by immunoblotting. Transfection of plasmid DNA were performed with ViaFect™ Transfection Reagent according to the manufacturer's instructions when cells were grown to 80% confluency. For primary adipocyte, transfection was performed when cells were differentiated into mature adipocytes.

## Histology analysis

BAT and iWAT tissue were dissected and fixed in 4% neutral-buffered paraformaldehyde (PFA) overnight at 4 °C and dehydrated through sequential ethanol washes, embedded in paraffin, and cut into 4-μm sections. Sections were stained by Hematoxylin and Eosin. For immunohistochemistry, paraffin-embedded sections of BAT and iWAT were probed with antibodies against UCP1 (Fitzgerald, 70R-UR001) or Tom20 (Proteintech,11802-1-AP). The Stained tissues were then scanned with Pannoramic MIDI (3DHISTECH) and analyzed by Case-Viewer2.4. For transmission electron microscopy, iWAT or primary adipocyte were fixed with 2.5% glutaraldehyde (pH7.4) overnight at 4 °C, and then postfixed with 2.0% osmium tetroxide for 1 h at RT. The samples were dehydrated through a graded series of ethanol and embedded in Spurr resin. Thin sections were stained with uranyl acetate and lead citrate and examined with a JEM-1010 electron microscope.

## Lipidomics analysis

For the lipidomic analysis[72], total lipids were extracted from iWAT of *Pnpla7$^{Tg}$* and *Pnpla7$^{AKO}$* mice. In brief, iWAT collected from *Pnpla7$^{Tg}$* and *Pnpla7$^{AKO}$* mice and instantly frozen in liquid nitrogen. Tissues were then lysed with 1 mL of methanol containing internal standards, and

then mixed with 1 mL chloroform and vortexed for 20 s. Subsequently, 400 μL water were added and again for 20 s vortex. The hydrophobic layer was collected and freeze-dried. The lyophilized powder was redissolved in 30 μl organic solvent (Chloroform/Methanol = 2/1) by 30 s vortex, then added 60 μl organic solvent (Acetonitrile/Isopropanpl/H2O = 65/30/5). After centrifugation, the lipid extract was injected onto a Kinetex C18 column (100 × 2.1 mm, 2.6 μm, Phenomenex) and BEH HILIC (100 × 2.1 mm, 1.7 μm, Waters) coupled to an QTRAP 6500 Plus mass spectrometers (SCIEX, Chromos, Singapore) with Internal Standards Kit for Lipidyzer™. Mass spectrometry analysis was carried out with the dynamic multiple reaction monitoring (MRM) mode. To control instrument drift, the original peak area was standardized to an internal standard for each lipid class. All samples were analyzed at the Metabolomics and Systems Biology Laboratory, School of Life Sciences, Fudan University.

## Quantification of mtDNA copy number

Mouse iWAT and primary adipocyte was digested in 500 μl DNA extraction buffer (10 mM Tris-HCl (pH 8.0), 1% SDS, 50 mM EDTA, 100 μg/mL proteinase K) overnight at 55 °C. After digestion, genomic DNA was extracted using a phenol/chloroform method as previously described[73]. The concentration and quality of genomic DNA were measured and subjected to further real-time PCR analysis. To quantify the number of mitochondria DNA (mtDNA), we used primers specific for the mitochondrial cytochrome c oxidase subunit 2 (COX2) gene and normalized to genomic DNA by the amplification of the FAS nuclear gene. The sequences of the primers used in this study were listed in Supplementary Table S1.

## Oxygen consumption rate measurements

OCR were measured using an extracellular flux analyzer (XFe24, Seahorse Bioscience). SVF cells isolated from iWAT were plated in an XF24-well microplate (Seahorse Bioscience) and differentiated into white or beige adipocytes, respectively, as described above. A Seahorse XF Cell Mito Stress test was performed by OCR measurement at 37 °C using an XF24 Analyser (Seahorse Bioscience) in accordance with the manufactures' instructions. One micrometer oligomycin, 2 μM FCCP, and 0.5 μM rotenone/antimycin were delivered to detect the uncoupled respiration, maximal respiration, and non-mitochondrial respiration, respectively.

## Lipolysis assay

The procedure for lipolysis assay was essentially the same as described[74]. Briefly, for ex vivo studies, iWAT isolated from 10-week-old male mice were cut into 1-3 mm and then washed with PBS followed by Opti-MEM with 2% fatty acid-free BSA for three times. Tissue pieces were incubated in 500 μL Opti-MEM containing 2% fatty acid-free BSA with or without 10 μM isoproterenol (ISO) and 10 μM IBMX as indicated. Ten microliter of medium was withdrawn at the indicated time points and used for the assay. Glycerol content of conditioned medium was determined with the Free Glycerol Determination Kit (Sigma). For in vivo studies[69], after 4 h fasting, mice were injected intraperitoneally with CL 316,243 (1 mg/kg). Blood was obtained from retro-orbital plexus of the mice before and 6 h after CL 316, 243 injected. The TAG, NEFA, and glycerol level in the serum was measured as above.

## Mitochondrial isolation and protein quantification

For mitochondria isolation from mouse adipose tissue[75]. Subcutaneous iWAT was digested with isolation buffer (0.01 M Tris-Hcl, 0.25 M source) supplemented with protease inhibitor cocktail (Roche), 1 mM PMSF, and 1 mM Na$_3$VO$_4$. The mixture was homogenized with a Dounce Tissue Grinder followed by layering the homogenize with same volume of medium containing (0.01 M Tris-HCl, 0.34 M source), and then the suspension was centrifuged twice at $700 \times g$ for 10 min at

4 °C. The resulting supernatants were collected and transferred to a new tube for another centrifugation at 10,000 × $g$ for 10 min at 4 °C and the resulted supernatants were discarded. The pellet containing the mitochondria was washed twice with isolation buffer and centrifuged at 10,000 × $g$ for 15 min at 4 °C. The mitochondria pallet was resuspended with RIPA buffer and mitochondrial protein concentrations were measured using the BCA Protein Assay Kit (Thermo Scientific, Rockford, IL, USA).

## Subcellular fractionation and MAM purification

For subcellular fractions purification[76]. 30 × 150 mm dishes of 3T3-L1 cells were homogenized with a Teflon potter in Isolation Buffer (225 mM mannitol, 75 mM sucrose, 1% BSA, 0.5 mM EGTA, and 30 mM Tris-HCl, pH 7.4). The cellular debris and nucleus were removed with two centrifugations at 740 × $g$ for 5 min. A small volume of supernatant known as the homogenate fraction, was collected. Crude mitochondria were collected by centrifugation at 9,000 × $g$ for 10 min, and the pellet was resuspended in mitochondria buffer (MB) (250 mM mannitol, 5 mM HEPES, and 0.5 mM EGTA, pH 7.4). The supernatant was conserved for ER purification. Pure mitochondria and MAM fractions were obtained from the crude mitochondria fraction with a Percoll medium centrifugation at 95,000 × $g$ for 30 min in a SW40 rotor (Beckman). Pure mitochondria (pM) at the bottom of the tube were collected, washed twice by centrifugation at 6300 × $g$ for 10 min, and resuspended in RIPA buffer. MAMs were collected from a white band in the middle of the tube and diluted in MB followed by centrifugation at 6300 × $g$ for 10 min to remove mitochondrial contamination. Finally, MAM was pelleted with a 1-h centrifugation at 100,000 × $g$ in a 70Ti rotor (Beckman), and resuspended in MB. The ER was purified by two centrifugations at 20,000 × $g$ and 100,000 × $g$ for 30 min and 1 h, respectively. Finally, the ER pellet was washed once in MB with a 10 min centrifugation at 9000 × $g$ and resuspended in RIPA buffer.

## Immunofluorescent staining

3T3-L1 cells were seeded on glass coverslips in 12-well plates and differentiated into mature adipocyte and then infected with Ad-control or Ad-mCherry-PNPLA7. After 48 h post infection, the ER and mitochondria were stained using ER-Tracker™ (Invitrogen, E34250) and Mito-Tracker™ (Invitrogen, M22426) for 20 min, the nuclei were stained using Hoechst 34580 (0.5 µg/mL) (Invitrogen, H21486). For FACL-4 staining, the cells were washed 3 times with PBS and fixed with 3.8% PFA in PBS for 20 min, permeabilized with 0.4% TritonX-100 for 10 min at RT. Permeabilized cells were blocked with 5% normal donkey serum for 1 h at RT. Subsequently, cells were incubated with PBS containing 1% BSA with mouse anti-FACL-4 antibody (Santa Cruz, sc-365230) overnight at 4 °C. Donkey anti-mouse antibody Alexa Fluor 488 (Invitrogen, A-21202) were used to conjugated anti-FACL-4-probe. The images were captured using Olympus FV1200 confocal microscope with a ×63/1.4 oil objective and analyzed by FV10-ASW software (4.0).

## mito-Keima assay

The mito-Keima lentivirus was kindly gifted by Dr. Quan Chen[77]. Wild type and PNPLA7 KO adipocytes were transduced with the mito-Keima lentiviruses. The indicated cells were plated onto confocal dishes and subjected to DMSO or CCCP (10 µM) treatment for 12 h after differentiating into mature adipocytes. The mito-Keima fluorescent signal from 568-nm laser excitation was depicted in red, and the signal from 488-nm laser excitation was in green. The fluorescence intensities of 568-nm and 488-nm excited mito-Keima signals were measured by Image J software, and the 561-nm:488-nm ratios were calculated.

## Parkin localization assay

Differentiated adipocytes from iWAT of wild type, *Pnpla7*$^{Tg}$, or PNPLA7 knockout mice were transfected with EGFP-Parkin for 48 h, after that,

cells were pretreated with CL316,243 at 1 µM or vehicle for 30 min. Subsequently, the cells were treated with CCCP at 10 µM for 3 h. Mitochondria were stained with Mito-Tracker™ (Invitrogen, M22426) for 20 min. The images were captured using Olympus FV1200 confocal microscope with a ×63/1.4 oil objective and analyzed by FV10-ASW software (4.0). Parkin recruitment to the mitochondria was determined on the basis of the merged images of EGFP-Parkin (488 nm) and Mito-tracker (568 nm) within a cell. For each experimental group, 20 pictures were randomly chosen from three independent experiments. Parkin protein recruitment to the mitochondria was analyzed by examining at least 200 cells that were randomly selected from each experimental group.

## Co-immunoprecipitation

For the co-immunoprecipitation assay, iWAT tissue and HEK293T cells were harvested and lysed with IP buffer containing 50 mM Tris-HCl (pH7.5);1 mM EGTA;1 mM EDTA; 1%Triton X-100; 150 mM NaCl; 2 mM DTT;100 µM PMSF, and complete protease inhibitor cocktail (Roche, 11697498001). One milligram of protein was immunoprecipitated with indicated antibodies or mouse IgG (Santa Cruz, sc-2025) overnight at 4 °C, and 30 µl protein A/G beads (Bimake, B23202) were added for an additional 2 h. The immuno-precipitates were washed with ice-cold lysis buffer followed by immunoblotting with the indicated antibodies. IPKine HRP Goat Anti-Mouse IgG LCS (Abbkine, A25012-1), which avoids the heavy chain, was used as the secondary antibody.

For the interaction analysis of PNPLA7 and Parkin after PKA activation, mice were injected with CL 316,243 (1 mg/kg) for 30 min while the primary adipocytes were treated with CL 316,243 (1 µM) for 30 min. Next, the subcutaneous iWAT tissue or primary adipocytes were harvested with IP buffer for cthe o-immunoprecipitation assay.

For the phosphorylated Parkin detection, mice were injected with CL 316,243 (1 mg/kg) for 30 min or primary adipocytes were treated with CL 316,243 (1 µM) for 30 min, with or without 1 µM H89 pretreatment for 1 h. Phosphorylated Parkin was detected by immunoprecipitation using the phospho-PKA substrate antibody, followed by immunoblot using the anti-Parkin antibody (CST, A25012-1).

## Adenovirus production and delivery

Adenovirus of PNPLA7 and PNPLA7$^{S983A}$ were generated and purified in our previous studies[43]. The primary adipocytes were isolated from iWAT of PNPLA7 knockout mice were infected with 6 × 10⁹ recombinant adenovirus particles of Ad-vector or Ad-PNPLA7 and Ad-PNPLA7$^{S983A}$ for 48 h, respectively. And then primary adipocytes were subjected to CCCP (10 µM) treatment for the indicated hours and cells were collected for subsequent experiments.

## Western blot analysis

Tissues or cultured cells were harvested and lysed in modified RIPA buffer supplemented with proteinase inhibitor cocktail (Roche), 5 mM NaF, 1 mM Na$_3$VO$_4$, 1 mM PMSF. The protein concentration was determined with BCA Protein Assay Kit (Thermo Scientific, Rockford, IL, USA), and an equivalent amount of protein was subjected to SDS-PAGE gel electrophoresis and transferred to pre-activated PVDF membranes (Millipore, Corp, USA). The membranes were blocked with 5% milk, and then incubated with the indicated primary antibodies, followed by HRP-conjugated secondary antibody. Protein was visualized using Super-Signal ECL (Tanon™ High-sig ECL Western Blotting Substrate). Signals were captured by a ChemiDoc XRS+ system (Bio-Rad), and the images were processed using Image Lab (version 4.1). The antibodies used in this study were listed in Supplementary Table S2.

To analyze Parkin E3 ligase activity, cell lysates were prepared in the modified buffer (pH 8.0) consisting of 20 mM Tris, pH 8.0, 150 mM NaCl, 0.5 % NP-40, 1 mM EDTA and 10 mM NEM then subjected to Western blot analysis as described.

## RNA isolation and quantitative real-time PCR

Total RNA was extracted from tissues or cells with TRIzol reagent (Invitrogen) according to the manufacturer's instructions. RNA concentrations were measured on a Nanodrop 1000 spectrophotometer (Thermo-Fisher). One microgram RNA was reverse transcribed with a high-capacity cDNA reverse transcription kit (Thermo Scientific, USA) and subjected to RT-PCR by using SYBR green mixture (Roche) with Light Cycler 480 II Sequence Detection System (Roche, Basel, Switzerland) according to the manufacturer's protocol. All qPCR data were analyzed using a standard curve and normalized to 36B4. The sequences of the specific primers used in this study were listed in Supplementary Table S1.

## Adipose tissue and serum analysis

Lipids were extracted from adipose tissue (20–25 mg) using the method of Folch and Lees[78]. The levels of TAG, cholesteryl ester, and NEFA were measured using enzymatic assays (Infinity and Wako Inc.) according to the manufacturer's instructions and normalized to sample weight. Blood was obtained from the eyeballs of the mice and clotted at RT for 1 h. The samples were then centrifuged (10 min, $1500 \times g$, 4 °C) and the supernatants were collected and stored at −80 °C until use. Serum levels of TAG, cholesteryl ester, and NEFA were measured using the same method according to the manufacturer's instructions.

## AAV8-Adiponectin-shParkin, AAV–Parkin^R274W and Ad-Adiponectin-Pnpla7 production, purification, and injection

GPAAV8-adipo-mcherry-5′mir30-scramble-3′mir30-WPRE and GPAAV8-adipo-mcherry-5′mir30-shParkin-mir30-WPRE were produced and purified by Genomeditech (Shanghai) Co.Ltd using these sequences (sh-Scramble-F:GAT CTG TTC TCC GAA CGT GTC ACG TTT CAA GAG AAC GTG ACA CGT TCG GAG AAT TTT TTC, sh-Scramble-R: AAT TGA AAA AAT TCT CCG AAC GTG TCA CGT TCT CTT GAA ACG TGA CAC GTT CGG AGA ACA; sh-Parkin-F: CGC GTC TGA CCA GCT GCG TGT GAT TTC TCG AGA AAT CAC ACG CAG CTG GTC AGG; sh-Parkin-R: GAT CCC TGA CCA GCT GCG TGT GAT TTC TCG AGA AAT CAC ACG CAG CTG GTC AGA). AAVGP2/9-CMV-MCS-EF1-ZsGreen1-WPRE and AAVGP2/9-CMV-Mouse-Parkin^R274W-EF1-ZsGreen1-WPRE were produced and purified by Genomeditech (Shanghai) Co. Ltd using these sequences (F: GCG AAT TCG AAG TAT ACC TCG AGG CCA CCA TGA TAG TGT TTG TCA GGT TCA; R: CGA TCG CAG ATC CTT GGA TCC CTA CAC GTC AAA CCA GTG ATC TCC C). pADV-Adiponectin-Pnpla7 and pADV-Adiponectin-mCherry were generated and purified by OBiO Technology (Shanghai) Corp. Ltd. For Adeno-Associated Virus (AAV) injections to knockdown Parkin and overexpress Parkin^R274W experiment in vivo, 4-week-old male mice were injected orthotopically with $2.0 \times 10^{11}$ vector genomes (VG) of AAV8-adipo-shParkin virus or control AAV8-adipo-shScramble virus at multiple sites in the iWAT of Pnpla7^Tg mice and littermate control mice. A dose of $5 \times 10^{10}$ VG of AAV–Parkin^R274W and AAV-vector was injected into the iWAT at multiple sites of Pnpla7^Tg mice and littermate control mice. After 28 days under normal chew diet, these mice were exposure to a cold environment (6 °C) for 7 days and sacrificed after cold stimulation. The subcutaneous WAT was collected for further analysis. For the generation of Ad-Adiponectin-Pnpla7 mice, 8-week-old male C57BL/6 J mice were injected orthotopically with $5.0 \times 10^{8}$ pfu of Ad-Adipo-mCherry or Ad-Adipo-Pnpla7 adenovirus at multiple sites in the subcutaneous iWAT and fed with normal chow diet for 4 days and subsequently transferred to 6 °C for 7 days to facilitate further investigation.

## Statistics and reproducibility

Quantifications of immunoblotting were performed by NIH ImageJ (Fiji). The indirect calorimetry analysis was completed on the CalR2 (https://calrapp.org)[71]. No statistical methods were used to predetermine sample size. Statistical analysis was performed by GraphPad Prism 9. No animals or data points were excluded from the analyses, and results were presented as mean ± SEM of at least three independent experiments as indicated in the figure legends. A two-tailed Student's t test was used for calculating the statistical significance of two groups. Statistical results were considered significant when $p < 0.05$. Experiments on mice were performed in a blinded fashion with random division based on genotype or ear tag number. Imaging and histology experiments were also performed in a blinded fashion. No data points were excluded from the study. The data were assumed to follow a continuous normal distribution.

## Reporting summary

Further information on research design is available in the Nature Portfolio Reporting Summary linked to this article.

## Data availability

The data that support the findings of this study are available within the paper and Supplementary Information. Source data for Figs. 1–7 and Supplementary Figs. 1–9 is provided as Source Data file and may be obtained from the corresponding authors upon request. Source data are provided with this paper.

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

## Acknowledgements

We thank Drs. Tong-Jin Zhao, Quan-Chen, Qi-Chen, Zheng-ji Gan and members of John Zhong Li's laboratory for help discussion and technical assistance. Drs. Boon Tin Chua, Yu-guan Shi critical reading of the manuscript. This work was supported by grants from the National Natural Science Foundation of China (32130050 and 92357307 to J.Z.L., 32201064 to X.Z., 82370873 and 82170838 to Q.W., 82330104 to A.H.G.), the National Key R&D Program of China (2024 YFA 1802803), and the Jiangsu Provincial Innovation Team Program Foundation to J.Z.L.

## Author contributions

Q.W. and J.Z.L. conceptualized and supervised this study. X.T.J. and X.Z. contributed to study design and performed most experiments. T.Z., Y.X., M.H., C.P.L., Y.H., H.Y.W., J.J., L.E.C., Y.Z.W., and N.W. contributed to the animal experiment. L.J.F. and H.T. contributed to plasmid construction. H.F., Q.S.C., Q.W.L., C.L., H.R.T., Y.S.C., X.X.K., H.M.S., A.H.G., H.L., H.W.Z., Q.W., and J.Z.L. participated in the result discussion. X.T.J., X.Z., Q.W., and J.Z.L. contributed to the discussion and wrote the manuscript. All authors approved the final manuscript. Q.W. and J.Z.L. are the guarantors of this work and, as such, had full access to all the data in the study and took responsibility for the integrity of the data and the accuracy of the data analysis.

## Competing interests

The authors declare no competing interests

## Additional information

¹The Key Laboratory of Rare Metabolic Disease, Department of Molecular Biology and Biochemistry, Jiangsu Key Laboratory of Molecular Targets and Intervention for Metabolic Diseases, The Key Laboratory of Targeted Intervention of Cardiovascular Disease, Collaborative Innovation Center for Cardiovascular Disease Translational Medicine, Nanjing Medical University, Nanjing, Jiangsu 211166, China. ²Institute of Human Stem Cells, General Hospital of Ningxia Medical University, Yinchuan, Ningxia, China. ³State Key Laboratory of Genetic Engineering, School of Life Sciences, Human Phenome Institute, Zhangjiang Fudan International Innovation Center, metabolomics and Systems Biology Laboratory at Shanghai International Centre for Molecular Phenomics, Zhongshan Hospital, Fudan University, Shanghai, China. ⁴Department of Physiology and Pathophysiology, Tianjin Medical University, Tianjin, China. ⁵State Key

Laboratory of Genetic Engineering, Shanghai Key Laboratory of Metabolic Remodeling and Health, Institute of Metabolism and Integrative Biology, School of Life Sciences, Fudan University, Shanghai, China. [6]Faculty of Health Sciences, University of Macau, Macau, China. [7]Department of General Surgery, The First affiliated Hospital of Nanjing Medical University, Nanjing, Jiangsu, China. [8]Department of Endocrinology, The First affiliated Hospital of Nanjing Medical University, Nanjing, Jiangsu, China. [9]Department of Endocrinology, The affiliated Huaian No.1 People's Hospital of Nanjing Medical University, Northern Jiangsu Institute of Clinical Medicine, Huaian, Jiangsu, China. [10]Tianjian Laboratory of Advanced Biomedical Sciences, Institute of Advanced Biomedical Sciences, Zhengzhou University, Zhengzhou, Henan, China. [11]These authors contributed equally: Xuetao Ji, Xu Zhang. ✉e-mail: wqian@njmu.edu.cn; lizhong@njmu.edu.cn

