## [Transparent Peer Review file · Nature Communications]

PNPLA7 mediates Parkin-mitochondrial recruitment in adipose tissue for mitophagy and inhibits browning

Corresponding Author: Professor John Li

Version 0:

Reviewer comments:

Reviewer #1

(Remarks to the Author)

The study by Ji et al. reports a series of studies to show that PNPLA7 is a relevant molecular actor in the control of adipose browning mediated by Parkin. The manuscript contains an impressive number of experiments to develop this research and establish the role of PNPLA7 and mechanisms of action. The authors propose a novel mechanism of control of browning based on PNPLA7 and acting through the Parkin-mediated control of mitophagy.

- Regarding originality, the involvement of PNPLA7 in adipose tissue mitophagy and adipose browning has been reported previously by Hirabayashi et al. (reference 44 in the manuscript) *in vivo*. A difference with the current manuscript is the more extensive and mechanistic analysis on this phenomenon here and provision of mechanistic insights, however, in Ref 44 authors postulated that the effect of PNPLA7 was not cell autonomous whereas the current manuscript shows cell autonomous actions. This discrepancy should be addressed more specifically in Discussion.

- The manuscript is largely based on the use of two genetically modified mouse models to generate gain- (transgenic aP2-driven over-expression of PNPLA7) and loss- (adiponectin-mediated CRE-LOX system invalidation of PNPLA7) of function, obtaining reciprocal results in the distinct experiments and parameters analyzed. The difference in the promoters used for the models may introduce some distortions. The aP2 promoter is known to drive expression in cells other than adipocytes, making it important for the authors to show no alterations in PNPLA7 in brain regions. Adipose browning is strongly controlled centrally, and changes can be indirect if PNPLA7 expression is altered in the brain.

- Moreover, there is a conceptual aspect relevant to the models used to study browning. The loss- and gain-of-function models used here are expected to act upon differentiated adipocytes. Thus, all observations related to increased or decreased browning in response to more or less abundance of PNPLA7 in adipocytes assume "transdifferentiation" processes in the adipocytes themselves. Although transdifferentiation is recognized to occur in the context of adipose browning, it is not necessarily the most relevant process, especially in the context of increased browning (rather than whitening of existing beige/brown cells). In such cases, the differentiation of precursor cells, not mature adipocytes, plays a key role. The overall experimental strategy should be framed within this context.

- For *in vitro* gain- and loss-of-function experiments, the authors use primary cultures from the aforementioned mouse models in most cases. This approach may be somewhat misleading for analyzing the specific role of PNPLA7 in several experiments (especially in the aP2-driven over-expression model) as the cells, originating during the embryonic period, may have indirect alterations associated with PNPLA7 loss or over-expression that occurred during previous mouse development. Besides showing the extent of loss and over-expression *in vitro* in the primary cultures, data should be confirmed with knockdown and/or over-expression strategies using wild-type cells and existing methods via interfering RNA and viral-driven overexpression of PNPLA7.

- Additionally, regarding cell cultures, the authors should at least justify the use of non-adipocyte cell models in some parts of the manuscript. Cell specificities cannot be ruled out a priori in relation to the subcellular localization of the molecular actors analyzed.

- The analyses provided in multiple sections of the manuscript lack quantitative assessment of the data and rely on the interpretation of images from immunoblots or other assays, often with just three replicates (extensively in Figure 5, for example). While the images in many experiments were considerably convincing, at key points, the provision of quantitative data with means and statistical deviation is necessary. Otherwise, it is incorrect to make statements about "significant" changes at several places in the manuscript.

- The results indicating a lack of effect of PNPLA7 in BAT (despite down-regulation in response to cold and beta3-adrenergic stimulus, like iWAT) deserve a more precise discussion, especially considering reference 37, which reports a role for the Parkin system in the control of BAT mitophagy and thermogenesis. Moreover, it is inconsistent that the authors

use only BAT for their subcellular assays.

- The manuscript requires a thorough revision for English syntax and wording. Some sentences contain writing mistakes that hinder proper reading. For some mistakes, it is unclear whether they are merely syntax errors or actual incorrect statements (e.g., second page of the Discussion, “many transcriptional regulators as well as various proteins such as PPAR γ , PRDM16, and PGC-1 α are....” None of these mentioned are secreted factors; the last sentence of the first page of the Discussion: what does “metabolic tissues” mean?). This should be revised throughout the manuscript to prevent misperceptions. In some cases, the references cited do not support the statements made (e.g., among the references 11-17 at the end of the first paragraph on the second page of the Introduction, or references 23-27 at the end of the second page of the Introduction, several do not strictly correspond to the previous sentences).
- A minor point is that the pages are not numbered, which made it unnecessarily laborious for this reviewer to quote specific places for required amendments.

Reviewer #2

(Remarks to the Author)

The manuscript of Xuetao Ji et al. described the role of PNPLA7 in adipocyte function with a focus on thermogenic adipocytes. The authors found that

- 1) PNPLA7 is regulated during thermogenic activity in adipocytes
- 2) PNPLA7 overexpression in adipose tissue is detrimental in acute cold challenge
- 3) Adipose tissue-KO mice show improved acute cold tolerance and UCP1 expression
- 4) PNPLA7 influences mitochondrial function and degradation

The authors gave a significant amount of novel insights into PNPLA7 biology and the influence on mitochondrial biology in adipocytes. Many of the results are very interesting and novel and an important contribution to the field. However, the manuscript has some major limitations.

Major:

1) The authors study a lysophospholipid processing enzyme, but so not show any data regarding this previously described function of the enzyme. The authors must show if there is an impact or not of their overexpression and knockout models with regard to the lipidomic landscape, in particular LPC/PC, LPE/PE. Many mitochondrial phenotypes, which they see, might be the impact of disturbed phospholipid landscape of the mitochondrial membranes or lipidated proteins (e.g. LC3) and the effects related to mitophagy may be subsequent results of the lipidomic changes in membranes and lipidated proteins. This questions whether the obtained the results summarized as bullet points 4 are really well and sufficiently addressed. I would like to know the opinions from the authors regarding this point, as they are experts in PNPLA7 biology to explain why they did not show these important control measurements.

2) The acute cold challenge is not a good marker for brown fat function. Acute cold leads mostly to shivering and any effect on body temperature might be a secondary effect of this. Brown fat function should be addressed using metabolic chambers and injection of CL316,243 after acclimatization at different housing temperatures as used in common protocols (e.g. acclimatization at RT or cold, then for few hours to thermoneutral state with a following CL injection and monitoring for several hours) addressing brown fat function. This questions whether the obtained the results summarized as bullet points 2+3 are currently analyzed in a state-of-the-art fashion.

Minor:

- 1) The authors should reduce overstatements (e.g. “dramatically repressed”)
- 2) The Seahorse data should not be normalized on protein, but e.g. on cell count of fixated cells after the measurements e.g. with DAPI – especially if you overexpress proteins or interfere with autophagy/mitophagy, which might change the protein amount per cell.

Reviewer #3

(Remarks to the Author)

Ji et al. investigate the role of PNPLA7, a lysophospholipase previously associated with lipid metabolism and lipid droplet dynamics, in regulating PINK1/Parkin-mediated mitophagy during the browning of inguinal white adipose tissue (iWAT). The authors demonstrate that PNPLA7 inhibits iWAT browning by promoting mitophagy under conditions such as cold exposure or β 3-adrenergic receptor stimulation. Overexpression of Pnpla7 using the aP2/Fabp4 promoter increases mitophagy and prevents browning, whereas, conversely adipocyte-specific knockout inhibits mitophagy and enhances browning. While the mechanistical data appears sound, including mutation-based immunoprecipitation experiments, the mouse data have major limitations. In the end, the evidence for a role of mitophagy via PNPLA7-PINK1/parkin crosstalk in adipose browning is unconvincing.

Major points:

1. Using aP2 for the overexpression is heavily confounded by the fact that aP2/Fabp4 is no longer accepted for adipocyte studies as it is also highly expressed in macrophages and endothelial cells. Therefore, the in vivo data regarding the overexpression are not useful. Performing the Overexpression by utilizing an Adiponectin-driven expression system would

make the data reliable.

2. Rectal temperature is not a thermogenesis-specific readout. It can also be influenced by various factors, including physical activity, stress, and circadian rhythms. The authors need to perform indirect calorimetry and measure non-shivering thermogenesis directly as described in PMID: 21177944.
3. Using a small field of view is unacceptable for looking at browning of iWAT as the tissue so heterogenous – an entire section of the fat pad is needed.
4. What is the phenotype of the floxed mice at thermoneutrality? I find it awkward that the authors use room temperature as a control as for mice, this is already cold (but it does not seem so here).
5. Fig. 3k,l: from which housing temperature were these cells? How does connect to the temperature dependency of the phenotype?
6. What is the explanation that only browning but not brown fat is affected? This does not make any sense, especially as the effects can be introduced even in COS7 cells.
7. It would be important to have some co-localization studies of mitochondrial markers such as Tom20 with autophagy markers like LC3. Also assessing the degradation of p62 could prove more evidence for increased activity of mitophagy. In addition, lysosomal inhibitors could be used to measure the accumulation of mitochondrial material in autophagosomes. This would directly define mitophagy flux rather than the mere presence of autophagosomes, which could indicate more or less autophagy taking place.
8. If Pnpla7 binds to Parkin during β -adrenergic signaling, then it would make sense to confirm this theory by overexpressing Pnpla7 and inhibiting the activity/Knocking down Parkin, to confirm that the metabolic effects are mediated through changes in mitophagy.

Minor Points:

9. A spellcheck is required
10. Please include the weights of the tissues
11. Please attach all uncropped original western blots with legends in the supplement
12. Why did the authors do single lane western blots if they performed most of the experiments 3 times? Why not load the samples together on a gel, when possible. This would increase the robustness of the data
13. Figure 1c + Ext Figure 2e: When comparing the BAT in these two figures, the Cold activation of Ucp1 protein appears to be very strong in 1c, but just minimal in Ext Figure 2e, similarly for PGC1a, how can this be such a big difference between 2 similarly treated groups? Furthermore, it seems as if there is almost no downregulation of Pnpla7 during cold in Ext Figure 2e, which should happen according to 1c.
14. It seems that there are 2 bands for some of the Pnpla7 blots, is there an unspecific band? It appears that in some cases the upper band is responding to the expected changes (overexpression) and in other cases the lower band (knockdown)
15. Comparing 3c and 3i, it seems that the increase of TOM20 is not as prominent in the WB as suggested by the IHC. Could the authors quantify the WB?
16. Figures 3i and j: It appears that only some of the OXPHOS complexes are down- or upregulated, respectively. How do the authors explain these differences? Also MFN1 and 2 are upregulated in graph i) while they are not in graph j), how can there be such a major difference?
17. Figure 3k: How can the baseline OCR of the Pnpla7TG adipocytes be lower if the changes are supposed to occur only after induction of cold/activation of the β -adrenergic pathway? Although a smaller samples size was used in Figure 3d, under RT conditions there seems to be no change in mtDNA, supporting the notion, that there is no prominent change in mitophagy during basal conditions.
18. Figure 5g: the Calnexin housekeeper is the same blot, please correct this
19. Figure 5b: It appears that Pink1 expression under RT is also higher in the overexpression group, although the authors state no visible changes.
20. Ext Figure 6f: After 6h CCCP there is no restoration of the Ser65 poly-ubiquitination after, when the FL protein or the mutant is given. It is only visible in the valinomycin treatment. Furthermore, only Tom20 is restored in some cases, but mfn2 is not restored after 6h of either CCCP or valinomycin

Reviewer #4

(Remarks to the Author)

Version 1:

Reviewer comments:

Reviewer #1

(Remarks to the Author)

In my opinion, the revised manuscript by Li and colleagues reasonably addresses my concerns and questions raised during the review of the original version. The addition of key experiments (e.g., adenoviral vector-mediated intervention to complement genetic-based intervention) has strengthened the consistency of the conclusions. In light of the relevance of these new data, it is necessary that the information in the Figs 1R and 2R provided in the rebuttal report (expression of

PNPLA7 in brain and hypothalamus in Tg mice, experiments in mice injected with Adenoviral vectors with adiponectin-drive Pnpla7) were incorporated to the manuscript, at least as supplemental and the results and conclusions of these experiments should be explicitly described in the text of the manuscript.

Some of the results remain somewhat surprising (e.g., the lack of effects in BAT itself despite marked common regulation in BAT and WAT in response to cold), but the analysis appears consistent enough to support the observations. However, in this sense, the heading of the first section of the Results should be modified. What is shown in Fig1 is not only effect on subcutaneous adipose tissue but a consistent overall effect both in subcutaneous WAT and in BAT.

The writing of the manuscript has also been clearly improved, but minor mistakes remain.

Reviewer #2

(Remarks to the Author)

The manuscript by Ji et al. has been largely revised and improved. Open questions remain especially regarding the tissue specific effect of their observations as the beige adipocyte play only a minor part in whole body energy metabolism of a mouse. The effects described e.g. in the new calorimetry data show that it is sufficient to impact the whole body metabolism. Yet, the phenotype is very moderate and the effects seen do not explain any cold intolerance of the mouse models as indicated by the acute cold challenge. Again, acute cold intolerance is overall not a phenotype of thermogenic adipocytes and especially not beige adipocytes, but is most of the times related to shivering or uncontrolled heat loss (affections of the dermal barrier? Uncontrolled tail heat loss and low vasoconstriction?) and yet an unexplained phenotype of the manuscript.

However, the manuscript shows its strengths in the connection of in vivo and in vitro data with a novel function for PNPLA7 independent of its (presumably low) lysophospholipase activity and thus, represents an important addition to the field. I suggest to consider to even omit the cold exposure data and just to show the calorimetry data (or putting the cold data to the supplement) as it raises more questions than it really answers and should not be used or repeated by others to study thermogenic adipose tissue function.

There are some minor concerns to address:

- Please reduce the tone of the manuscript. Statements like "To validate this intriguing observation [...]" should be used more neutral. Please let the reader decide whether they find it intriguing or not. Please revise the manuscript according to a scientific and neutral tone.
- Downregulation/upregulation are terms which can only be used in specific circumstances like a time course with same samples which get repetitively measured. Please use terms like higher/lower levels or similar.

Reviewer #3

(Remarks to the Author)

The authors have provided important revisions based on the reviewers' input. Some of the concerning animal data has been repeated with a more suitable model and the authors have provided further necessary in vitro experiments and assays to strengthen their claims. This particularly includes brain- and adipocyte differentiation-related effects of PNPLA7, the specific lack of effects on BAT, the aP2 overexpression model and additional co-localization experiments. Furthermore, the authors have edited the majority of overstatements and adjusted their overinterpretations. Some points remain unclear and require revision:

1. The authors designed an adenoviral delivery system with an adiponectin promoter, to overexpress PNPLA7. Here, they do show that under cold stimulus, PNPLA7 overexpression impacts browning makers such as UCP1 protein expression. However, these animals were not in the metabolic cages, so it remains unclear if the effects on Ucp1 protein are actually changing thermogenesis (Rectal temperature was used again as a measure of thermogenesis, which it is not). An important question remains about Parkin, namely how it is affected within this adiponectin overexpression model. Also in the immunohistochemistry, TOM20 appears highly reduced with cold and PNPLA7 overexpression, whereas in the immunoblot this is not the case?
2. As rectal temperature is not a thermogenesis-specific readout the authors have now used indirect metabolic cages at least in their other genetic models of PNPLA7, however they must not normalize VO₂ per kg body weight (see PMID: 34489606 and PMID: 22205519). Otherwise, the energy expenditure data cannot be interpreted. Also, they need to include all other indirect calorimetry data, including activity and food intake.
3. The authors have added further experiments and important data to the manuscript, further characterizing the interplay of PNPLA7 and Parkin during β -adrenergic signaling. However, this experiment is clearly missing the AAV-shParkin Control group

Reviewer #4

(Remarks to the Author)

Version 2:

Reviewer comments:

Reviewer #3

(Remarks to the Author)

Thank you for addressing my comments. I have no further questions at this stage.

Reviewer #4

(Remarks to the Author)

First of all, we would like to express our gratitude to the editor and reviewers for their time, positive comments and helpful advices. Specifically, we have addressed these comments and concerns in a point-by-point manner as follows:

Reviewer #1:

Comment 1. The study by Ji et al. reports a series of studies to show that PNPLA7 is a relevant molecular actor in the control of adipose browning mediated by Parkin. The manuscript contains an impressive number of experiments to develop this research and establish the role of PNPLA7 and mechanisms of action. The authors propose a novel mechanism of control of browning based on PNPLA7 and acting through the Parkin-mediated control of mitophagy.

- Regarding originality, the involvement of PNPLA7 in adipose tissue mitophagy and adipose browning has been reported previously by Hirabayashi et al. (reference 44 in the manuscript) *in vivo*. A difference with the current manuscript is the more extensive and mechanistic analysis on this phenomenon here and provision of mechanistic insights, however, in Ref 44 authors postulated that the effect of PNPLA7 was not cell autonomous whereas the current manuscript showed cell autonomous actions. This discrepancy should be addressed more specifically in Discussion.

Response: We greatly appreciate the Reviewer's thorough review and invaluable suggestions. In previous study by Hirabayashi et al. (*Cell Rep. 2023; 423(2):111940*), PNPLA7 null mice exhibited browning phenotype in WAT while BAT remains unaffected. The authors proposed that browning in WAT is mediated by a non-cell autonomous mechanism based on the observation where by primary adipocytes derived from the null mice displayed normal level of LD accumulation without browning phenotype during differentiation.

In our study, we observed similar phenotype in primary adipocytes when differentiated under the same basal condition. However, considering that the induction of browning *in vivo* requires external stimuli, such as cold exposure, therefore, we further subjected SVF cells to Triiodothyronine (T3) treatment. T3 is a potent hormone used for brown adipocyte differentiation. Here, SVF cells are induced to beige adipocytes. Interestingly, overexpression of PNPLA7 represses oxygen consumption rates in the mitochondrial respiration assay (Figure 3k), while ablation of PNPLA7 results in enhanced oxygen consumption (Figure 3l). In addition, these *in vitro* differentiated adipocytes induced from SVF of *Pnpla7^{AKO}* exhibits less mitophagy when treated with CCCP or Valinomycin (Figure 4a-4g). Conversely, differentiated adipocytes induced from SVF of *Pnpla7^{Tg}* displays enhanced mitophagy with the same treatment (Supplementary Figure 6a-6g). Taken together, these comprehensive results demonstrate that PNPLA7 regulates browning and mitophagy in a cell-autonomous manner. As suggested by the reviewer, we have included this important point in the revised manuscript. Please refer to Page 26, line 10-22 and Page 27, line 1-6.

Comment 2. The manuscript is largely based on the use of two genetically modified mouse models to generate gain-(transgenic aP2-driven over-expression of PNPLA7) and loss- (adiponectin-mediated CRE-LOX system invalidation of PNPLA7) of function, obtaining reciprocal results in the distinct experiments and parameters analyzed. The difference in the promoters used for the models may introduce some distortions. The aP2 promoter is known to drive expression in cells other than adipocytes, making it important for the authors to show no alterations in PNPLA7 in brain regions. Adipose browning is strongly controlled centrally, and changes can be indirect if PNPLA7 expression is altered in the brain.

Response: Thank you for the comment. As advised, we have further evaluated PNPLA7 expression pattern in the brain of *Pnpla7^{Tg}* mice by Western blot analysis. Our data show that there is no obvious difference detected between PNPLA7 expression in

the brain of *Pnpla7^{Tg}* mice and wild-type mice (Figure R1a, b). In addition, given that the hypothalamus and medulla oblongata play an important role in central regulation of thermogenesis (*Cell Metab* 2014; 20(2): 346-358; *Cell Metab* 2017; 25(2): 322-334), we also examined PNPLA7 expression pattern in these two tissues. Similarly, no detectable differences were found in these regions (Figure R1c, d). Combine together, these data show that aP2-promoter driven overexpression strategy has no effect on PNPLA7 expression in the brain of *Pnpla7^{Tg}* mice.

Figure R1

Figure R1: (a) Representative Immunoblot results of PNPLA7 protein level in the brain of 10-week-old wild-type control (WT) and *Pnpla7^{Tg}* mice under normal chow diet (n=4/group). (b) Representative Immunoblot results of PNPLA7 protein level in iWAT of 10-week-old wild-type control (WT) and *Pnpla7^{Tg}* mice under normal chow diet (n=4/group). (c) Representative Immunoblot results of PNPLA7 protein level in hypothalamus of 10-week-old wild-type control (WT) and *Pnpla7^{Tg}* mice under normal chow diet (n=4/group). (d) Representative Immunoblot results of PNPLA7 protein level in medulla oblongata of 10-week-old wild-type control (WT) and *Pnpla7^{Tg}* mice under normal chow diet (n=4/group)

Furthermore, we specifically introduced Ad-Adiponectin-*Pnpla7* into iWAT of wild type mice via orthotopic injection, and these mice exhibited impaired iWAT browning capacity following cold stimulation (Figure R2). In these animals, a modest increase in fat mass (Figure R2a, b) with significant reduction in (1) UCP1 and Tom20 staining intensity (Figure R2c), (2) thermogenic and mitochondrial proteins expression (Figure R2d) and (3) mtDNA copy number (Figure R2f). These data indicate that the impaired browning phenotype in iWAT of *Pnpla7*^{Tg} mice is probably due to PNPLA7 overexpression in adipose tissue rather than off-site effect at the central nerve system.

Figure R2

Figure R2: 8-week-old male C57BL/6J mice were injected with recombinant adenovirus of Ad-Adiponectin-*mCherry* or Ad-Adiponectin-*Pnpla7* into the subcutaneous inguinal white adipose tissue (iWAT) and fed with normal chow diet. Subsequently, these mice were exposed to 6°C for 7 days and sacrificed. (a) Weight

ratio of iWAT harvested from Ad-Adiponectin-*mCherry* (Ad-control) and PNPLA7 overexpressing (Ad-Pnpla7) mice under room temperature and cold exposure. (RT: n = 5/group; Cold: n=6). Data are presented as mean \pm SEM. **p < 0.01 (two-tailed Student's t test). (b) TAG levels of iWAT obtained from control and PNPLA7 overexpressing male mice under room temperature and prolonged cold exposure at 6 °C for 7 days under normal chow diet (RT: n=5/group; Cold: n=6/group). Data are presented as mean \pm SEM. *p < 0.05 (two-tailed Student's t test). (c) Representative H&E, UCP1 and TOM20 immunohistochemical staining images of iWAT sections from control and PNPLA7 overexpressing mice. (d) Representative Immunoblot results of the indicated proteins in iWAT harvested from control and PNPLA7 overexpressing mice under room temperature and cold exposure. (e) Quantitative PCR of indicated mRNA levels of genes involved in mitochondrial biogenesis, mitochondrial OXPHOS, and lipolysis or fatty acid β -oxidation in iWAT of control and PNPLA7 overexpressing mice after prolonged cold exposure at 6 °C for 7 days under normal chow diet (RT: n = 5/group; cold: n=6/group). (f) Relative mtDNA content of iWAT harvested from control and PNPLA7 overexpressing mice under room temperature and cold exposure. (RT: n = 5/group; Cold: n=6). Data are presented as mean \pm SEM. (two-tailed Student's t-test).

Comment 3. Moreover, there is a conceptual aspect relevant to the models used to study browning. The loss- and gain-of-function models used here are expected to act upon differentiated adipocytes. Thus, all observations related to increased or decreased browning in response to more or less abundance of PNPLA7 in adipocytes assume “trans-differentiation” processes in the adipocytes themselves. Although trans-differentiation is recognized to occur in the context of adipose browning, it is not necessarily the most relevant process, especially in the context of increased browning (rather than whitening of existing beige/brown cells). In such cases, the differentiation of precursor cells, not mature adipocytes, plays a key role. The overall experimental strategy should be framed within this context.

Response: We agree with the reviewer's point of view that differentiation of precursor cells may play a key role in the context of adipose browning. However, in our study, both overexpression and deficiency of PNPLA7 were only found in mature adipocytes in *Pnpla7^{Tg}* or *Pnpla7^{AKO}* mice, respectively, therefore, differentiation of precursor cells has no role in the remarkable change observed in the browning ability of the white fat following cold stimulation. The browning phenotype that we observed is indeed a trans-differentiation effect (Figure 2). In addition, when we manipulated the expression of E3 ligase Parkin in iWAT of *Pnpla7^{Tg}* mice, the mature adipocytes preserved the ability to undergo browning (Figure 5h~5l, Supplementary Figure 8a~8e). Collectively, these findings indicate that "trans-differentiation" plays an important role in the browning process, which is consistent with the other studies (*Am J Physiol Endocrinol Metab.* 2010;298(6): E1244-1253; *Nat Rev Drug Discov.* 2016;15(6): 405-424; *Adv Sci.* 2020; 7(12): 1903366; *Cell Rep.* 2023; 423(2):111940). These new data are presented and further described in the revised manuscript. Please refer to Page 11, line 3-20, Page 19, line 19-22 and Page 20, line 1-15.

Comment 4, For in vitro gain- and loss-of-function experiments, the authors use primary cultures from the aforementioned mouse models in most cases. This approach may be somewhat misleading for analyzing the specific role of PNPLA7 in several experiments (especially in the aP2-driven over-expression model) as the cells, originating during the embryonic period, may have indirect alterations associated with PNPLA7 loss or over-expression that occurred during previous mouse development. Besides showing the extent of loss and over-expression in vitro in the primary cultures, data should be confirmed with knockdown and/or over-expression strategies using wild-type cells and existing methods via interfering RNA and viral-driven overexpression of PNPLA7.

Response: We would like to thank the reviewer for the experimental insight. As advised, using adenoviral vectors, we overexpress or knockdown PNPLA7 in wild-type stromal vascular fraction (SVF) cells and differentiated them into beige adipocytes

using T3. Here, we found that neither overexpression nor knockdown of PNPLA7 impacts the expression of the differentiation marker protein PPAR γ or the mitochondrial biogenesis-related gene PGC-1 α (Figure R3a, b). PNPLA7 specifically modulates the expression of the thermogenic gene UCP1 solely under conditions that are conducive to cellular browning (Figure R3a, b). These findings are consistent with observations made in SVF derived from *Pnpla7^{Tg}* and *Pnpla7^{AKO}* mice.

Figure R3

Figure R3: (a) Representative Western blot analysis of PNPLA7, PGC1-a, UCP1, PPAR γ and Calnexin proteins expression in mature adipocytes. Preadipocytes were isolated from *Pnpla7^{Flox}* mice and differentiated into mature adipocytes with or without T3. Ad-Cre was used to knockdown PNPLA7 before preadipocytes were differentiated into mature adipocytes. (b) Representative Western blot analysis of PNPLA7, PGC1a, UCP1, PPAR γ and Calnexin proteins expression in mature adipocytes. Preadipocytes were isolated from wild type mice and differentiated into mature adipocytes with or without T3. Ad-Pnpla7 was used to overexpression PNPLA7 before preadipocytes were differentiated into mature adipocytes.

Comment 5, Additionally, regarding cell cultures, the authors should at least justify the use of non-adipocyte cell models in some parts of the manuscript. Cell specificities cannot be ruled out a priori in relation to the subcellular localization of the molecular actors analyzed.

Response: Thank you for your advice. As suggested, we performed subcellular

fractionation using 3T3-L1 adipocytes. Consistent with the previous findings, we found that PNPLA7 is not only localized to the endoplasmic reticulum but also on the mitochondria-associated membrane (MAM) (Figure 6a, b). This is further evidenced by the colocalization of PNPLA7 with FAACL4, a representative marker of MAM in the immunofluorescence images (Figure 6 c). The subcellular localization of PNPLA7 provides spatial possibility for its interaction with Parkin. These new data are presented as Figure 6a~6d and further described in the revised manuscript. Please refer to Page 21, line 1-14.

Comment 6, The analyses provided in multiple sections of the manuscript lack quantitative assessment of the data and rely on the interpretation of images from immunoblots or other assays, often with just three replicates (extensively in Figure 5, for example). While the images in many experiments were considerably convincing, at key points, the provision of quantitative data with means and statistical deviation is necessary. Otherwise, it is incorrect to make statements about “significant” changes at several places in the manuscript.

Response: We agree with the reviewer’s concern on the quantitative assessment of the data. At least three independent repetitive experiments were performed to draw the conclusions presented in this work. We have revisited the images provided in the manuscript and conducted quantitative analysis. The supplementary information has been added to the Figures or Supplementary Materials accordingly.

Comment 7, The results indicating a lack of effect of PNPLA7 in BAT (despite down-regulation in response to cold and beta3-adrenergic stimulus, like iWAT) deserve a more precise discussion, especially considering reference 37, which reports a role for the Parkin system in the control of BAT mitophagy and thermogenesis. Moreover, it is inconsistent that the authors use only BAT for their subcellular assays.

Response: We would like to thank the reviewer for the comment. In this work, the role of PNPLA7 in BAT appears to be lacking based on multiple functional tests

performed on mice after chronic cold exposure (Supplementary Figure 1f-j, 2c-f). The phenotypic differences between iWAT and BAT may arise due to the fundamental difference in mitophagy in WAT during browning, exhibiting cellular specificity. Recent studies have demonstrated that Parkin-mediated mitophagy plays a crucial role in maintaining beige fat cells (*Sci Signal.* 2018; 11(527): eaap8526). Adipose tissue-specific Parkin knockout preserves browning and ameliorate obesity and blood glucose disorders induced by high-fat diet (HFD) feeding or aging (*Nat Commun.* 2022;13(1):6661; *J Physiol Biochem.* 2024; 80(1): 41-51). Interestingly, specific knockout of Parkin in BAT using UCP1-Cre shows minimal change in these phenotypes (*Nat Commun.* 2022;13(1):6661), indicating that the function of Parkin-mediated mitophagy in the regulation of metabolic homeostasis occurs mainly in the iWAT rather than BAT, which deserves further investigation. Since PNPLA7 involvement in the regulation of iWAT browning is dependent on Parkin, it is therefore reasonable to conclude that PNPLA7 has little effect on BAT after chronic cold exposure. We have discussed this in the revised manuscripts in Page 9, line 15-22 and Page 10, line 1-5 , Page 28, 5-22 and Page 29, 1-6.

Further, as suggested, subcellular fractionation was performed on 3T3-L1 adipocytes to determine PNPLA7 subcellular localization. Consistent with the previous findings, PNPLA7 is found to localize to the endoplasmic reticulum and mitochondria-associated membrane (MAM) (Figure 6a). These new data are presented as Figure 6a and further described in the revised manuscript. Please refer to Page 21, line 1-7.

Comment 8, The manuscript requires a thorough revision for English syntax and wording. Some sentences contain writing mistakes that hinder proper reading. For some mistakes, it is unclear whether they are merely syntax errors or actual incorrect statements (e.g., second page of the Discussion, “many transcriptional regulators as well as various proteins such as PPAR γ , PRDM16, and PGC-1 α are....” None of these mentioned are secreted factors; the last sentence of the first page of the Discussion: what does “metabolic tissues” mean?). This should be revised throughout the manuscript to prevent misperceptions. In some cases, the references cited do not

support the statements made (e.g., among the references 11-17 at the end of the first paragraph on the second page of the Introduction, or references 23-27 at the end of the second page of the Introduction, several do not strictly correspond to the previous sentences).

Response: Thank you for your thorough review. We have made comprehensive revisions to the manuscript wording. We have also re-examined the strict correspondence between the references and the content of the article. All changes are marked in **red font**. Some local references have also been updated, such as:

“As in humans, this browning process has been suggested to have anti-obesity and anti-diabetic benefits and represents a promising strategy to counteract obesity and metabolic dysfunctions ¹⁰⁻¹⁷.” replaced by “This browning process has been suggested to have anti-obesity and anti-diabetic benefits and represents a promising strategy to counteract obesity and metabolic dysfunctions ¹⁰⁻¹⁷.” and we have updated the references 11-17.

“Emerging evidences suggest that mitophagy plays a key role in brown and beige adipocyte mitochondrial homeostasis during cold adaptation”. The statement reflects what we intended to express, and we have also updated the list of references namely, ref 23-24.

Comment 9, A minor point is that the pages are not numbered, which made it unnecessarily laborious for this reviewer to quote specific places for required amendments.

Response: Thank you for your suggestion. We have added page number as footer to the document.

Reviewer #2:

Major:

Comment 1, The authors study a lysophospholipid processing enzyme, but so not show any data regarding this previously described function of the enzyme. The authors must show if there is an impact or not of their overexpression and knockout models with regard to the lipidomic landscape, in particular LPC/PC, LPE/PE. Many mitochondrial phenotypes, which they see, might be the impact of disturbed phospholipid landscape of the mitochondrial membranes or lipidated proteins (e.g. LC3) and the effects related to mitophagy may be subsequent results of the lipidomic changes in membranes and lipidated proteins. This questions whether the obtained the results summarized as bullet points 4 are really well and sufficiently addressed. I would like to know the opinions from the authors regarding this point, as they are experts in PNPLA7 biology to explain why they did not show these important control measurements.

Response: We appreciate your valuable suggestion. We did a survey on the lipidomic landscape, including LPC/PC and LPE/PE in the adipose tissues isolated from *Pnpla7^{Tg}* and *Pnpla7^{AKO}* mice both under room temperature and cold exposure. The results show no significant difference in *Pnpla7^{Tg}* or *Pnpla7^{AKO}* mice and their littermate control mice. As such, these data were not presented previously. In the revision, we have added these data in Supplementary Figure 5. Our findings indicate that neither overexpression nor knockdown of *Pnpla7* in adipose tissues significantly affects the total levels of LPC/PC, LPE/PE, or LPS/PS. Similarly, there are also no significant changes in the different species of PC/LPC and PE/LPE, which is consistent with the overall phospholipid and lysophospholipid levels. These observations suggest that PNPLA7 probably retains low lysophospholipase activity in the adipose tissue. However, PNPLA7 protein *per se* has regulatory function. Additionally, we found that upon treatment with CCCP or Valinomycin, overexpression of both wild-type and lysopholipase-inactive PNPLA7 (S983A) mutant proteins in PNPLA7-deficient adipocytes are sufficient to restore the ubiquitination of MFN2, TOM20, and pSer65-Ub, which are downstream targets of Parkin E3 ligase (Supplementary Figure 7). With

no significant changes to the lipid class analyzed, these results suggest that PNPLA7's role in promoting mitophagy is independent of its lysophospholipase activity. We have discussed these findings on Page 12, line 20-22 ; Page 13, line 1-12 ; Page 30, line 10-22 and Page 31, line 1-2.

Comment 2, The acute cold challenge is not a good marker for brown fat function. Acute cold leads mostly to shivering and any effect on body temperature might be a secondary effect of this. Brown fat function should be addressed using metabolic chambers and injection of CL316,243 after acclimatization at different housing temperatures as used in common protocols (e.g. acclimatization at RT or cold, then for few hours to thermoneutral state with a following CL injection and monitoring for several hours) addressing brown fat function. This question whether the obtained the results summarized as bullet points 2+3 are currently analyzed in a state-of-the-art fashion.

Response: We appreciate your valuable suggestion. Based on your recommendation, we performed indirect calorimetry experiment and recorded the oxygen consumption and energy expenditure of both male *Pnpla7^{Tg}* and *Pnpla7^{AKO}* mice. The results showed that both *Pnpla7^{Tg}* and *Pnpla7^{AKO}* mice exhibited similar oxygen consumption and energy expenditure as the wild type littermates under normal condition. Interestingly, after CL316,243 treatment, a reduction in oxygen consumption and energy expenditure were recorded for *Pnpla7^{Tg}* mice while an increased in oxygen consumption and energy expenditure were observed for *Pnpla7^{AKO}* mice. These new data are presented as Figure 2b,2c,2j,2k and further described in the revised manuscript. Please refer to Page 9, line 12-15 and Page 11, line 9-11.

Minor:

Comment 1, The authors should reduce overstatements (e.g. “dramatically repressed”)

Response: Thank you for the advice. We have meticulously reviewed the manuscript and have carefully minimized any overstatements in the revised version.

Comment 2, The Seahorse data should not be normalized on protein, but e.g. on cell count of fixated cells after the measurements e.g. with DAPI – especially if you overexpress proteins or interfere with autophagy/mitophagy, which might change the protein amount per cell.

Response: Thank you for your insightful suggestion. We have revised our protocol and performed the experiment as follows: SVF cells were isolated from iWAT of *Pnpla7^{Tg}* and *Pnpla7^{AKO}* mice and differentiated into beige adipocytes *in vitro*. Subsequently, 10⁵ cells per well were seeded into the seahorse plate. The cells were fixed after seahorse measurement and DAPI staining was used to perform cell count. The revised seahorse data was normalized to cell number and the updated data is presented in Figure 3k, l. Please refer to Page 14, line 16-22 and Page 15, line 1-8.

Reviewer 3:

Major points:

Comment 1, Using aP2 for the overexpression is heavily confounded by the fact that aP2/Fabp4 is no longer accepted for adipocyte studies as it is also highly expressed in macrophages and endothelial cells. Therefore, the in vivo data regarding the overexpression are not useful. Performing the Overexpression by utilizing an Adiponectin-driven expression system would make the data reliable.

Response: Thank you for your insightful suggestion. In this manuscript, in addition to using aP2-promoter driven strategy to overexpress PNPLA7 in adipose tissue where we discovered the downregulation of iWAT browning in *Pnpla7^{Tg}* (Figure 2a-2h), we also generated adipose tissue-specific PNPLA7 knockout mice by using the Adiponectin-driven expression strategy. In contrast to the phenotype observed in *Pnpla7^{Tg}* mice, *Pnpla7^{AKO}* mice shows enhanced browning capacity (Figure 2i-2p).

To further address your concern, as advised, we constructed an adenoviral delivery system with adiponectin promoter and utilized orthotopic injection method to overexpress PNPLA7 proteins specifically in iWAT. Subsequently, the phenotype of iWAT under both room temperature and cold exposure were analyzed. Here, PNPLA7 overexpression in adipocytes has no effect on iWAT under room temperature. Under chronic cold exposure, mice with PNPLA7 overexpression display an increase in inguinal fat pad weight and TAG content compared to Adiponectin-*mCherry* control group (Figure R2a, b). In addition, H&E and immunohistochemical staining revealed greater number of large lipid droplets and fewer UCP1⁺ and TOM20⁺ beige adipocytes in iWAT of PNPLA7 overexpressing mice (Figure R2c). Furthermore, Western blotting showed the absence of UCP1 and lower mitochondrial-associated proteins were induced in PNPLA7 overexpressing iWAT (Figure R2d). No difference in mRNA levels was observed between PNPLA7 overexpressing and control iWAT (Figure R2e). Last but not least, mitochondrial DNA significantly increase with cold exposure indicating browning in progress in control iWAT. This phenotype is absent in PNPLA7 overexpressing iWAT (Figure R2f). Thus, using adiponectin promoter to drive PNPLA7

overexpression in adipose tissue further consolidate the role of PNPLA7 in iWAT. As observed in *aP2-Pnpla7^{Tg}* mice, overexpressing PNPLA7 using adiponectin promoter in adipocytes inhibits iWAT browning during cold exposure. Taken together, PNPLA7 plays a negative role in browning process when expose to browning stimuli.

Figure R2

Figure R2: 8-week-old male C57BL/6J mice were injected with recombinant adenovirus of Ad-Adiponectin-*mCherry* or Ad-Adiponectin-*Pnpla7* into the subcutaneous inguinal white adipose tissue (iWAT) and fed with normal chow diet. Subsequently, these mice were exposed to 6 °C for 7 days and sacrificed. (a) Weight ratio of iWAT harvested from control (Ad-Adiponectin-*mCherry*) and PNPLA7 overexpressing (Ad-Adiponectin-*Pnpla7*) mice under room temperature and cold exposure. (RT: n = 5/group; Cold: n=6). Data are presented as mean \pm SEM. **p < 0.01 (two-tailed Student's t test). (b) TAG levels of iWAT obtained from control and PNPLA7 overexpressing male mice under room temperature and prolonged cold

exposure at 6 °C for 7 days under normal chow diet (RT: n=5/group; Cold: n=6/group). Data are presented as mean ± SEM. *p < 0.05 (two-tailed Student's t test). (c) Representative H&E, UCP1 and TOM20 immunohistochemical staining images of iWAT sections from control and PNPLA7 overexpressing mice. (d) Representative Immunoblot results of the indicated proteins in iWAT harvested from control and PNPLA7 overexpressing mice under room temperature and cold exposure. (e) Quantitative PCR of indicated mRNA levels of genes involved in mitochondrial biogenesis, mitochondrial OXPHOS, and lipolysis or fatty acid β-oxidation in iWAT of control and PNPLA7 overexpressing mice after prolonged cold exposure at 6 °C for 7 days under normal chow diet (RT: n = 5/group; cold: n=6/group). (f) Relative mtDNA content of iWAT harvested from control and PNPLA7 overexpressing mice under room temperature and cold exposure. (RT: n = 5/group; Cold: n=6). Data are presented as mean ± SEM. (two-tailed Student's t-test).

Comment 2. Rectal temperature is not a thermogenesis-specific readout. It can also be influenced by various factors, including physical activity, stress, and circadian rhythms. The authors need to perform indirect calorimetry and measure non-shivering thermogenesis directly as described in PMID: 21177944.

Response : Thank you for your advice. Based on your recommendation, we performed indirect calorimetry experiment and recorded the oxygen consumption and energy expenditure of both male *Pnpla7^{Tg}* and *Pnpla7^{AKO}* mice. The results showed that both *Pnpla7^{Tg}* and *Pnpla7^{AKO}* mice exhibited similar oxygen consumption and energy expenditure to their wild type littermates under normal conditions. Interestingly, after CL316,243 treatment, a reduction in oxygen consumption and energy expenditure were recorded for *Pnpla7^{Tg}* mice while an increase in oxygen consumption and energy expenditure were observed for *Pnpla7^{AKO}* mice. These new data are presented in Figure 2b,2c,2j,2k and further described in the revised manuscript. Please refer to Page 9, line 12-15 and Page 11, line 9-11.

Comment 3. Using a small field of view is unacceptable for looking at browning of iWAT as the tissue so heterogenous – an entire section of the fat pad is needed.

Response: Thank you for your suggestion. As per your request, we have made the necessary revision. The field of view was widened for H&E and immunohistochemical staining sections performed on iWAT and BAT. All the sections of iWAT include preserved lymph nodes which is the main area where browning occurs. In addition, the entire section of the fat pad is provided in supplementary data. These new data are presented as Figure 2g,2o,3c,3g, 5j and Supplementary Figure 1f,1g,1k,2c,8c.

Comment 4. What is the phenotype of the floxed mice at thermoneutrality? I find it awkward that the authors use room temperature as a control as for mice, this is already cold (but it does not seem so here).

Response: Thank you for your insightful suggestion. The phenotype of *Pnpla7^{Tg}* and *Pnpla7^{AKO}* mice at thermoneutrality was analyzed. Firstly, there is no difference between the body weight of *Pnpla7^{Tg}* mice (Figure R4a) and *Pnpla7^{AKO}* mice (Figure R4f) when compared to their littermate control. Next, both *Pnpla7^{Tg}* and *Pnpla7^{AKO}* mice exhibit similar adiposity index (Figure R4b, 4g) and fat pad weight (Figure 4c, 4h), indicating PNPLA7 expression has no effect on the fat mass. In addition, H&E and UCP and TOM20 immunohistochemical staining (Figure R4d, 4i) as well as Western blotting of the indicated markers (Figure R4e, 4j) show no significant differences between *Pnpla7^{Tg}* and *Pnpla7^{AKO}* mice with their littermate control at thermoneutrality. Taken together, the results show that there is no phenotypical difference in *Pnpla7^{Tg}* and *Pnpla7^{AKO}* mice at thermoneutrality and room temperature, thus, the use of either as control for our model is feasible.

Figure R4

Figure R4: (a) Body weight analysis of control and *Pnpla7^{Tg}* mice after 30°C thermonutral challenge for 7 days under normal chow diet. (n=6/group. Data are presented as mean ± SEM. (b) Adiposity index analysis of control and *Pnpla7^{Tg}* mice after 30°C thermonutral challenge for 7 days under normal chow diet. (n=6/group. Data are presented as mean ± SEM. (c) Fat mass weight ratio analysis of control and *Pnpla7^{Tg}* mice after 30°C thermonutral challenge for 7 days under normal chow diet. (n=6/group). Data are presented as mean ± SEM. (d) Representative H&E and UCP1

immunohistochemical staining images of iWAT sections from control and *Pnpla7^{Tg}* male mice after 30°C thermonutral challenge for 7 days under normal chow diet (n=3/group). Scale bar=100 μm. (e) Representative Immunoblot results of the indicated proteins in iWAT of control and *Pnpla7^{Tg}* male mice after 30°C thermonutral challenge for 7 days under normal chow diet. (f) Body weight analysis of control and *Pnpla7^{AKO}* mice after 30°C thermonutral challenge for 7 days under normal chow diet. (n=6/group. Data are presented as mean ± SEM. (g) Adiposity index analysis of control and *Pnpla7^{AKO}* mice after 30°C thermonutral challenge for 7 days under normal chow diet. (n=6/group. Data are presented as mean ± SEM. (h) Fat mass weight ratio analysis of control and *Pnpla7^{AKO}* mice after 30°C thermonutral challenge for 7 days under normal chow diet. (n=6/group). Data are presented as mean ± SEM. (i) Representative H&E and UCP1 immunohistochemical staining images of iWAT sections from control and *Pnpla7^{AKO}* male mice after 30°C thermonutral challenge for 7 days under normal chow diet (n=3/group). Scale bar=100 μm. (j) Representative Immunoblot results of the indicated proteins in iWAT of control and *Pnpla7^{AKO}* male mice after 30°C thermonutral challenge for 7 days under normal chow diet.

Comment 5. Fig. 3k, l: from which housing temperature where these cells? How does connect to the temperature dependency of the phenotype?

Response: Thank you for your question. The SVF cells were isolated from inguinal WAT of 3-week-old *Pnpla7^{Tg}* and *Pnpla7^{AKO}* mice. These cells were cultured in DMEM/F-12 medium containing 10% fetal bovine serum (FBS) and 1% penicillin/streptomycin (P/S) at 37°C in an incubator with 5% CO₂ in the atmosphere. Notably, despite being cultured at 37°C, we induced browning with the addition of T3 into the differentiation medium to generate mature adipocytes. T3 serves as the stimulus to initiate browning process in the cellular model. This is parallel to the cold exposure and treatment with β₃-adrenergic receptor agonist CL316,243 which induce browning in mouse models. These findings suggest that PNPLA7 exerts a negative regulatory effect on browning both *in vitro* and *in vivo* when subjected to browning-inducing

stimuli.

Comment 6. What is the explanation that only browning but not brown fat is affected? This does not make any sense, especially as the effects can be introduced even in COS7 cells.

Response: We agree with the reviewer's concern on the role of PNPLA7 in BAT. In this work, the role of PNPLA7 in BAT appears to be lacking based on multiple functional tests performed on a large number of mice after chronic cold exposure (Supplementary Figure 1f-j, 2c-f). The phenotypic differences between iWAT and BAT may arise due to the fundamental difference in mitophagy in WAT during browning, exhibiting cellular specificity. Recent studies have demonstrated that Parkin-mediated mitophagy plays a crucial role in maintaining beige fat cells (Sci Signal. 2018; 11(527): eaap8526). Adipose tissue-specific Parkin knockout preserves browning and ameliorate obesity and blood glucose disorders induced by high-fat diet (HFD) feeding or aging (Nat Commun. 2022;13(1):6661; J Physiol Biochem. 2024; 80(1): 41-51). Interestingly, specific knockout of Parkin in BAT using UCP1-Cre shows minimal change in these phenotypes (Nat Commun. 2022;13(1):6661), indicating that the function of Parkin-mediated mitophagy in the regulation of metabolic homeostasis occurs mainly in the iWAT rather than BAT, which deserves further investigation. Since PNPLA7 involvement in the regulation of iWAT browning is dependent on Parkin, it is therefore reasonable to conclude that PNPLA7 has little effect on BAT after chronic cold exposure. We have discussed this in the revised manuscript in Page 9, line 15-22 ; Page 10, line 1-5; Page 28, 6-22 and Page 29, 1-6.

Here, we would like to clarify that the use of Cos7 cells was only intended for the PNPLA7 subcellular localization analysis and not functional experiments. To be more relevant with our studies, we further performed both subcellular fractionation and immunofluorescence staining analysis in 3T3-L1 adipocytes. The results show that PNPLA7 is not only localized to the endoplasmic reticulum but also anchored on the mitochondria-associated membrane (MAM) as we observed in Cos7 cells (Figure 6a-

c). These new data are presented as Figure 6a~6d and further described in the revised manuscript. Please refer to Page 21, line 1-14.

Comment 7. It would be important to have some co-localization studies of mitochondrial markers such as Tom20 with autophagy markers like LC3. Also assessing the degradation of p62 could prove more evidence for increased activity of mitophagy. In addition, lysosomal inhibitors could be used to measure the accumulation of mitochondrial material in autophagosomes. This would directly define mitophagy flux rather than the mere presence of autophagosomes, which could indicate more or less autophagy taking place.

Response: Thank you for your suggestion. Accordingly, we analyzed the co-localization of mitochondrial with autophagic markers LC3 in PNPLA7-overexpressed or -knockout primary adipocytes that were differentiated in vitro. The data show that overexpression of PNPLA7 increases LC3 puncta formation which colocalized with Tom20 after CCCP treatment, indicating enhanced autophagy and mitophagy. In contrast, although knockout of PNPLA7 reduces the formation of LC3 puncta, colocalization between Tom20 with LC3 remains, suggesting a reduced autophagy and mitophagy (Figure R5a, b). In addition, P62 protein expression did not displayed significant change, fulfilling its role as an autophagic indicator rather than mitophagy, as suggested by reviewer (Figure 5c and supplementary figure 7c). Please refer to Page 18, line 11-12.

Furthermore, we included mito-Keima assay, TEM, and the degradation of mtDNA and mitochondrial-associated proteins to support the role of PNPLA7 in regulating mitophagy flux. In mito-Keima experiment, after CCCP treatment, an increase and reduction in red fluorescence were observed for PNPLA7 overexpression and PNPLA7-deficient adipocytes, respectively. The data clearly indicate altered mitophagy flux (Figure 4a, b and supplementary figure 6a, 6b). Subsequently, upon quantification of the number of mitophagosomes per 100 mitochondria, our findings revealed that overexpression of PNPLA7 enhances the formation of mitophagosomes compared to control cells in the presence of CCCP. Conversely, in PNPLA7 deficient

cells, the induction of mitophagosomes is repressed by at least 50% (Figure 4c, d, and Supplementary Figure 6c, d). Furthermore, overexpressing PNPLA7 enhances, while knocking out PNPLA7 diminishes the degradation of mitochondrial DNA and mitochondrial-associated proteins in response to CCCP treatment (Figure 4e, f, g, and Supplementary Figure 6e, f, g). These results unequivocally demonstrate that PNPLA7 regulates mitophagy flux. Please refer to Page 15, line 12-22 and Page 16, line 1-17.

Figure R5

Figure R5: (a) Colocalization of GFP-LC3 with mitochondria in differentiated adipocytes. Differentiated adipocytes isolated from iWAT of wild-type control (WT) and PNPLA7 knockout (KO) male mice were infected with EGFP-LC3 for 48 h and then incubated with DMSO (vehicle) or CCCP (10 μ M) for 3 h. Mitochondria were stained with Tom20 (red), EGFP-LC3 translocation to mitochondria were analyzed by confocal microscopy. (n=3 biological replicates). Scale bar=20 μ m. (b) Colocalization of GFP-LC3 with mitochondria in differentiated adipocytes. Differentiated adipocytes isolated from iWAT of wild-type control (WT) and PNPLA7 overexpression (*Pnpla7^{Tg}*) male mice were infected with EGFP-LC3 for 48 h and then incubated with DMSO (vehicle) or CCCP (10 μ M) for 3 h. Mitochondria were stained with Tom20 (red), EGFP-LC3 translocation to mitochondria were analyzed by confocal microscopy. (n=3 biological replicates). Scale bar=20 μ m.

Comment 8. If *Pnpla7* binds to Parkin during β -adrenergic signaling, then it would

make sense to confirm this theory by overexpressing Pnpla7 and inhibiting the activity/Knocking down Parkin, to confirm that the metabolic effects are mediated through changes in mitophagy.

Response: Thank you for the suggestion. As advised, endogenous Parkin was knock down using Adeno-Associated Virus (AAV-shParkin) (Fig 5h). The results showed that Parkin deficiency indeed restored impaired browning capacity caused by PNPLA7 overexpression. These include the weight of iWAT (Fig 5i), mtDNA (Fig 5j), UCP1+ and TOM20+ beige adipocytes IHC intensity (Fig. 5k) in iWAT of *Pnpla7^{Tg}*. Furthermore, under cold exposure, the lowered expression of mitochondrial protein such as MFN1, MFN2, TOM20, UCP1, and OXPHOS in PNPLA7 overexpressing iWAT of *Pnpla7^{Tg}* were also restored with the knockdown of Parkin (Fig 5l). To highlight the importance of this collaboration between PNPLA7 and Parkin, we overexpressed E3 ligase inactive mutant Parkin^{R274W} in iWAT of *Pnpla7^{Tg}* mice to inhibit endogenous Parkin E3 ligase. Consistently, overexpression of Parkin^{R274W} recapitulates the effect of Parkin knockdown whereby the downregulated browning in iWAT was restored in *Pnpla7^{Tg}* after prolonged cold exposure (Supplementary Fig. 8a-e). These data clearly demonstrate that overexpression of dominant negative Parkin^{R274W} or knockdown of endogenous Parkin reverses PNPLA7 inhibition on iWAT browning. These new data are presented as Figure 5h~5l and Supplementary Fig 8a-e and further described in the revision. Please refer to Page19, line 19-22; Page 20, line 1-15.

Minor Points:

Comment 9. A spellcheck is required

Response: Thank you for the reminder. As advised, we have performed a thorough spellcheck in the revised manuscript.

Comment 10. Please include the weights of the tissues

Response: Thank you for your suggestion. We have supplemented the ratio of fat

pad weight to body weight in Figure 2e and Figure 2m in the revised manuscript. Please refer to Page 10, line 9 and Page 11, line 14.

Comment 11. Please attach all uncropped original western blots with legends in the supplement

Response: Thank you for your suggestion. As advised, we have attached all the original, uncropped Western blots with figure legends in the supplementary data.

Comment 12. Why did the authors do single lane western blots if they performed most of the experiments 3 times? Why not load the samples together on a gel, when possible, This would increase the robustness of the data.

Response: Thank you for your suggestion. We have loaded samples in triplicates on a single gel and quantified each band on the immunoblots (Figure 4f, 4g and supplementary figure 6f, 6g). These are included in the revised manuscript

Comment 13. Figure 1c + Ext Figure 2e: When comparing the BAT in these two figures, the Cold activation of Ucp1 protein appears to be very strong in 1c, but just minimal in Ext Figure 2e, similarly for PGC1a, how can this be such a big difference between 2 similarly treated groups? Furthermore, it seems as if there is almost no downregulation of Pnpla7 during cold in Ext Figure 2e, which should happen according to 1c.

Response: Thank you for your meticulous question and comments. We have quantified UCP1 protein levels in Figure 1c and Ext Figure 2e (Figure 1c + Supplementary Figure 2e in the revised manuscript). Specifically, the quantitative data indicate that UCP1 increased by ~3-4 times and 2.5 times in Figure 1c and Supplementary Figure 2e, respectively. The minor difference arises as we normalized the data to wild type mice in Figure 1c and *Pnpla7^{fllox/fllox}* control mice in Supplementary Figure 2e. The expression pattern of PGC1a is similar to UCP1, however, PGC1a did not display significant increase in BAT due to its high abundance at room temperature.

This is consistent with the result in Supplementary Figure 2e. For Figure 1c, only PGC1a in iWAT was presented. Last but not least, we quantified PNPLA7 protein levels in both Figure 1c and Supplementary Figure 2e. Consistently, upon cold induction, PNPLA7 expression reduced by ~50% in both figures.

Comment 14. It seems that there are 2 bands for some of the Pnpla7 blots, is there an unspecific band? It appears that in some cases the upper band is responding to the expected changes (overexpression) and in other cases the lower band (knockdown)

Response: Thank you for the question. There are indeed non-specific bands on Pnpla7 blots. As the PNPLA7 antibody is a homemade polyclonal antibody, therefore, two bands were observed for some of the Pnpla7 blots. Please note that only the lower band in PNPLA7 blots were quantitated in overexpression and knockdown conditions.

Comment 15. Comparing 3c and 3i, it seems that the increase of TOM20 is not as prominent in the WB as suggested by the IHC. Could the authors quantify the WB?

Response: Thank you for the question. As advised, we quantified the expression of TOM20 in Figure 3g and the result indicates that the expression of TOM20 in iWAT of *Pnpla7^{Tg}* mice reduces by 50% compared to the littermate control. This is consistent with immunohistochemistry data. We have updated TOM20 IHC image with a zoomed-in view in Figure 3c in the revised manuscript.

Comment 16. Figures 3i and j: It appears that only some of the OXPHOS complexes are down-or regulated, respectively. How do the authors explain these differences? Also MFN1 and 2 are upregulated in graph i) while they are not in graph j), how can there be such a major difference?

Response: Thank you for questions. The mitochondrial complexes are made of different subunits and their expression differ. Both ATP5A and UQCRC2 are usually expressed at higher level compared to the other three subunits especially NDUFB8 of

complex I. Therefore, during mitophagy, ATP5A and UQCRC2 is less likely to show significant change compared to the other three subunits.

We thank the reviewer for pointing out the problem of MFN1 and MFN2. We repeated the Western blot and updated the new blots in Figure 3j.

Comment 17. Figure 3k: How can the baseline OCR of the *Pnpla7*TG adipocytes be lower if the changes are supposed to occur only after induction of cold/activation of the b-adrenergic pathway? Although a smaller samples size was used in Figure 3d, under RT conditions there seems to be no change in mtDNA, supporting the notion, that there is no prominent change in mitophagy during basal conditions.

Response: Thank you for your valuable question. We would like to clarify that for Figure 3k we isolated SVF cells from iWAT of *Pnpla7^{Tg}* mice and used T3 to induce differentiation to beige adipocytes to mimic the response of adipose tissue when the animal is exposed to cold or when β -adrenergic pathway is activated. Mitochondrial functional analysis using Seahorse respirometry showed no significant difference observed in PNPLA7 overexpressing adipocytes compared to control cells in the absence of T3. This is consistent with our mice data under room temperature. However, upon Beigeing induction with T3, adipocytes with PNPLA7 overexpression exhibit very little change for basal, proton leak-linked and maximal cellular oxygen consumption rates (Fig. 3k) compared to control cells with T3. Conversely, ablation of PNPLA7 enhances mitochondrial spare respiratory capacity in mature primary beige adipocytes (Fig. 3l). Please refer to Page 14, line 16-22 and Page 15, line 1-8.

Comment 18. Figure 5g: the Calnexin housekeeper is the same blot, please correct this

Response: Thank you for the kind reminder. We have rectified the mistake in the revised manuscript.

Comment 19. Figure 5b: It appears that Pink1 expression under RT is also higher in

the overexpression group, although the authors state no visible changes.

Response: Thank you for the comment. The quantified data for the Immunoblot (Fig. 5b) indeed indicate that there is no significant increase in PINK1 protein in the total lysate prepared from iWAT.

Comment 20. Ext Figure 6f: After 6h CCCP there is no restoration of the Ser65 poly-ubiquitination after, when the FL protein or the mutant is given. It is only visible in the valinomycin treatment. Furthermore, only Tom20 is restored in some cases, but mfn2 is not restored after 6h of either CCCP or valinomycin

Response: We appreciate your valuable comments. we repeated the Western blotting and updated the new blot for pSer65 poly-ubiquitination in Supplementary Figure 7f. Due to Parkin-mediated proteasomal degradation, the degradation rate of ubiquitinated MFN2 appears to be the fastest, while TOM20 the slowest (J Cell Biol. 2010; 191(7): 1367-1380; Elife. 2018; 7: e32866). Therefore, TOM20 retains its ubiquitinated state for longer duration, whereas MFN2-ubiquitinated state is the shortest. In our cell model, we observed a significant restoration of pSer65 poly-ubiquitination and ubiquitination of MFN2 and TOM20 at 4 hours post-CCCP or valinomycin treatment. However, most of the ubiquitinated MFN2 is degraded by 6 hours post-treatment. Consequently, there is a relatively minor restoration of MFN2 levels compared to TOM20 at the 6-hour mark following CCCP or valinomycin treatment.

First of all, we would like to express our gratitude to the editor and reviewers for their time, positive comments and helpful advices. Specifically, we have addressed these comments and concerns in a point-by-point manner as follows:

Reviewer #1

Comment 1. In my opinion, the revised manuscript by Li and colleagues reasonably addresses my concerns and questions raised during the review of the original version. The addition of key experiments (e.g., adenoviral vector-mediated intervention to complement genetic-based intervention) has strengthened the consistency of the conclusions. In light of the relevance of these new data, it is necessary that the information in the Figs 1R and 2R provided in the rebuttal report (expression of PNPAL7 in brain and hypothalamus in Tg mice, experiments in mice injected with Adenoviral vectors with adiponectin-drive *Pnpla7*) were incorporated to the manuscript, at least as supplementary and the results and conclusions of these experiments should be explicitly described in the text of the manuscript.

Response: We greatly appreciate the thorough review and invaluable suggestions. The information in Figs 1R and 2R provided in the previous rebuttal report has been incorporated to the supplementary data. PNPLA7 protein expression in the brain, hypothalamus and medulla oblongata of *Pnpla7^{Tg}* and littermate control mice is shown in Supplementary Figure 1e~g. As advised, we administered Ad-Adiponectin-*Pnpla7* via orthotopic injection into the inguinal white adipose tissue (iWAT) of wild-type mice. Our data show that these mice exhibit reduced oxygen consumption and energy expenditure along with impaired iWAT browning capacity following prolonged cold exposure. These data are presented as Supplementary Figure 6a~m. In addition, we have explicitly described these new results in the revised manuscript. Please refer to Page 13, line 16-22 and Page 14, line 1-9.

Comment 2. Some of the results remain somewhat surprising (e.g., the lack of effects

in BAT itself despite marked common regulation in BAT and WAT in response to cold), but the analysis appears consistent enough to support the observations. However, in this sense, the heading of the first section of the Results should be modified. What is shown in Fig1 is not only effect on subcutaneous adipose tissue but a consistent overall effect both in subcutaneous WAT and in BAT.

Response: Thank you for your suggestion. As advised, we have revised the heading of figure 1 to "PNPLA7 is downregulated in adipose tissues during browning" Please refer to Page 7, line 6 and Page 54, line 9.

Comment 3. The writing of the manuscript has also been clearly improved, but minor mistakes remain.

Response: Thank you for the reminder. A thorough spellcheck has been conducted in the revised manuscript.

Reviewer #2

Comment 1. The manuscript by Ji et al. has been largely revised and improved. Open questions remain especially regarding the tissue specific effect of their observations as the beige adipocyte play only a minor part in whole body energy metabolism of a mouse. The effects described e.g. in the new calorimetry data show that it is sufficient to impact the whole body metabolism. Yet, the phenotype is very moderate and the effects seen do not explain any cold intolerance of the mouse models as indicated by the acute cold challenge. Again, acute cold intolerance is overall not a phenotype of thermogenic adipocytes and especially not beige adipocytes, but is most of the times related to shivering or uncontrolled heat loss (affections of the dermal barrier? Uncontrolled tail heat loss and low vasoconstriction?) and yet an unexplained phenotype of the manuscript.

However, the manuscript shows its strengths in the connection of in vivo and in vitro data with a novel function for PNPLA7 independent of its (presumably low)

lysophospholipase activity and thus, represents an important addition to the field. I suggest to consider to even omit the cold exposure data and just to show the calorimetry data (or putting the cold data to the supplement) as it raises more questions than it really answers and should not be used or repeated by others to study thermogenic adipose tissue function.

Response: We sincerely appreciate the reviewer's thoughtful and constructive feedback on our study. As advised, we have removed the acute cold exposure data from the revised manuscript to avoid overinterpretation. In the meantime, the indirect calorimetry data including oxygen consumption, energy expenditure, physical activity and food intake are shown in Figure 2a, 2b, 2h, 2i and Supplementary Figure 1h, 1i, 2c, 2d, 6a–d of the revised manuscript. The data description is found on Page 8, line 10-15; Page 9, line 19-22 and Page 10, line 1-4 as well as Page 13, line 16-22.

Minor

Comment 1. -Please reduce the tone of the manuscript. Statements like “To validate this intriguing observation [...]” should be used more neutral. Please let the reader decide whether they find it intriguing or not. Please revise the manuscript according to a scientific and neutral tone.

-Downregulation/upregulation are terms which can only be used in specific circumstances like a time course with same samples which get repetitively measured. Please use terms like higher/lower levels or similar.

Response: Thank you for the suggestion and advice. We have conducted a thorough review and revision to ensure more objective language and precise terminology.

Reviewer #3

The authors have provided important revisions based on the reviewers' input. Some of the concerning animal data has been repeated with a more suitable model and the authors have provided further necessary in vitro experiments and assays to strengthen their claims. This particularly includes brain- and adipocyte differentiation-related effects of PNPLA7, the specific lack of effects on BAT, the aP2 overexpression model

and additional co-localization experiments. Furthermore, the authors have edited the majority of overstatements and adjusted their overinterpretations. Some points remain unclear and require revision:

Comment 1. The authors designed an adenoviral delivery system with an adiponectin promoter, to overexpress PNPLA7. Here, they do show that under cold stimulus, PNPLA7 overexpression impacts browning makers such as UCP1 protein expression. However, these animals were not in the metabolic cages, so it remains unclear if the effects on Ucp1 protein are actually changing thermogenesis (Rectal temperature was used again as a measure of thermogenesis, which it is not). An important question remains about Parkin, namely how it is affected within this adiponectin overexpression model. Also in the immunohistochemistry, TOM20 appears highly reduced with cold and PNPLA7 overexpression, whereas in the immunoblot this is not the case?

Response: Thank you for the comment. As advised, we performed indirect calorimetry experiment on mice with PNPLA7 overexpression driven by adiponectin promoter. The oxygen consumption and energy expenditure data reveal that overexpression of PNPLA7 in iWAT reduces oxygen consumption and energy expenditure after CL 316,243 injection, without affecting physical activity and food intake. This indicates that reduction of UCP1 protein in iWAT may directly change thermogenesis. The data are presented in Supplementary Figure 6a-d. We agreed that rectal temperature is not suitable for thermogenesis measurement, thus, the readings were omitted from the acute cold exposure data in the revised manuscript to avoid overinterpretation as suggested. Please refer to Page 8, line 10-15; Page 9, line 19-22 and Page 10, line 1-4 as well as Page 13, line 16-22.

As depicted in Figure 5c and Supplementary Figure 8c, mitochondrial translocation of Parkin is positively regulated by PNPLA7. Further examination of *Ad-Pnpla7* mice model indicate that overexpression of PNPLA7 in iWAT promotes Parkin mitochondrial translocation. These data were presented in Supplementary Figure 8d-e. Please refer to Page 16, line 19-22 and Page 17, line 1-7.

Last but not least, we would like to thank the reviewer for addressing Tom20. We repeated the experiment and protein bands were quantified on the immunoblots. Upon cold induction, consistently ~50% reduction was observed for TOM20 proteins in PNPLA7 overexpression group compared to the control group. We have updated the new data in Supplementary Figure 6h in this revision.

Comment 2. As rectal temperature is not a thermogenesis-specific readout the authors have now used indirect metabolic cages at least in their other genetic models of PNPLA7, however they must not normalize VO₂ per kg body weight (see PMID: 34489606 and PMID: 22205519). Otherwise, the energy expenditure data cannot be interpreted. Also, they need to include all other indirect calorimetry data, including activity and food intake.

Response: We sincerely appreciate your professional advice. As suggested, the acute cold exposure data have been omitted from the revised manuscript to avoid overinterpretation. As advised, we conducted a regression-based analysis with ANCOVA (Analysis of Covariance) on the indirect metabolic cage data as described in PMID: 34489606 and PMID: 22205519. The result obtained from oxygen consumption, energy expenditure, physical activity and food intake are shown in Figure 2a,2b,2h,2i and Supplementary Figure 1h, 1i, 2c, 2d, 6a–d of the revised manuscript. Please also refer to Page 8, line 10-15; Page 9, line 19-22 and Page 10, line 1-4 as well as Page 13, line 16-22.

Comment 3. The authors have added further experiments and important data to the manuscript, further characterizing the interplay of PNPLA7 and Parkin during β -adrenergic signaling. However, this experiment is clearly missing the AAV-shParkin Control group

Response: Thank you for the suggestion. As advised, in the revision, we have included the data of AAV-shParkin control group in Figure 5h-l. Upon cold exposure, *Pnpla7^{Tg}* mice exhibited a marked impairment in browning capacity compared to littermate control mice. In our model, PNPLA7 mediates the phosphorylation and mitochondrial

translocation of Parkin. Expectedly, overexpression of PNPLA7 in *Pnpla7^{Tg}* mice enhances Parkin mitochondrial translocation, increases mitophagy and impairs browning process. Consequently, knock down of Parkin eliminates the difference in browning capacity between the *Pnpla7^{Tg}* and littermate control mice. This include mtDNA as well as UCP1 and mitochondrial associated protein levels. Consistent with Kajimura et al. (*Sci Signal* **11**, eaap8526 (2018)), no obvious difference on browning capacity was found in the wild type mice with Parkin knock down. Taken together, these data suggest that PNPLA7 activates Parkin mediated mitophagy by promoting Parkin's mitochondrial recruitment to enhance mitochondrial degradation, thereby impairing iWAT browning. Please refer to Page 18, line 12-22.

Reviewer #4:

Response: Thank you for your time, thoughtful comments and valuable advice.

REVIEWERS' COMMENTS

Reviewer #3 (Remarks to the Author):

Thank you for addressing my comments. I have no further questions at this stage.

Responses: We greatly thank Reviewer 3 for his/her highly valuable concerns, comments and suggestions, which largely help on generating this significantly improved version of manuscript!